# Calibration of hydrological models for ecologically-relevant streamflow predictions: a trade-off between performance and consistency?

Thibault Hallouin[1], Michael Bruen[1], and Fiachra E. O'Loughlin[1]

[1]UCD Dooge Centre for Water Resources Research, University College Dublin, Ireland.

**Correspondence:** Thibault Hallouin (th.thibault.hallouin@gmail.com)

**Abstract.** The ecological integrity of freshwater ecosystems is intimately linked to natural fluctuations in the river flow regime. In catchments with little human-induced hydro-morphological changes, existing hydrological models can be used to predict changes in local flow regime in order to assess whether its rivers remain a suitable living environment for endemic species. However, hydrological models are traditionally calibrated to give a good general fit between observed and simulated hydrographs, e.g. using objective functions such as the Nash-Sutcliffe, or the Kling-Gupta efficiencies. Much ecological research has shown that aquatic species respond to very specific characteristics of the hydrograph, whether magnitude, frequency, duration, timing, and rate of change of flow events. This study investigates the performance of specially developed, tailored, objective functions made of combinations of such specific streamflow characteristics found to be ecologically-relevant in previous eco-hydrological studies. These are compared with the more traditional objective functions based on the Kling-Gupta efficiency on a set of 33 Irish catchments. A split-sample test with a rolling-window procedure is applied to reduce the influence of variations between the calibration/evaluation periods on the conclusions. These tailored objective functions are shown to be marginally better suited to predict the targetted streamflow characteristics in terms of performance in evaluation; however, traditional objective functions are more robust considering both calibration and evaluation periods, and produce more consistent behavioural parameter sets, suggesting a trade-off between model performance and model consistency when predicting streamflow characteristics. Analysis of the objective function performances on a set of 165 streamflow characteristics revealed a general lack of versatility for objective functions with a strong focus on low flow conditions, especially in predicting high flow conditions. On the other hand, the Kling-Gupta efficiency applied to the square-root of flow values performs as well as two sets of tailored objective functions across the 165 streamflow characteristics. These findings suggest that traditional composite objective functions such as the Kling-Gupta efficiency may still be preferable over tailored objective functions for the prediction of streamflow characteristics, when robustness and consistency are important.

## 1 Introduction

River flow is the cornerstone of freshwater ecosystems, the ecological integrity of which relies on the natural fluctuations in the river flow regime (Poff et al., 1997). A long history of human alterations of river flow regime for water supply, irrigation, flood protection, or hydropower threatens water security and freshwater biodiversity in many regions of the world (Vörösmarty

et al., 2010). Richter et al. (1997) raised the overarching research question "How much water does a river need?". In order to quantify these needs and assess the effects of altered flow regime on freshwater ecology, many different hydrological indices have been used, whether they are referred to as streamflow characteristics (SFC) (Vis et al., 2015; Pool et al., 2017), ecologically relevant flow statistics (ERFS) (Caldwell et al., 2015), or indicators of hydrological alterations (IHA) (Richter et al., 1996).

These SFCs describe specific aspects of the river flow regime that can be extracted from the streamflow hydrograph, and they can be categorised on the basis of magnitude, frequency, rate of change, timing, and duration of high, average, and low flow events (Poff et al., 1997). Olden and Poff (2003) listed a range of such SFCs used to characterise river flow regime in relation to ecological species' preferences. The prediction of these SFCs at ungauged locations has historically being done using statistical analyses such as regional regression models that relate them to some climatic and physiographic descriptors

(e.g. Carlisle et al., 2011; Knight et al., 2014). However, these regression models are not well-suited for investigating water management or climate change scenarios because they often rely on long-term, stationary, descriptors. On the other hand, hydrological models can allow for such scenario analyses, and they produce simulated streamflow hydrographs from which the streamflow characteristics can be computed (e.g. Shrestha et al., 2014; Caldwell et al., 2015).

Most rainfall-runoff models used to predict these SFCs relevant for the stream ecology require calibration to determine their

effective model parameter values for the catchment of interest. The selection of the objective function(s) in the calibration process is of great importance for the quality of the predictions of SFCs (Vis et al., 2015; Kiesel et al., 2017; Pool et al., 2017). As demonstrated by Vis et al. (2015), equally performing parameter sets obtained through the usual calibration methods with the Nash-Sutcliffe efficiency (NSE) criterion (Nash and Sutcliffe, 1970) fitted to flows can yield very different performances when looking at the prediction of SFCs. This exposes the limitations of models in representing the entirety of real-world

processes in a catchment. Indeed, because of uncertainties in model structure, model forcing, and evaluation data (Beven, 2016), the identification of a single perfect parameter set is usually unachievable (Beven, 2006), and in practice trade-offs are required between modelling different aspects of the hydrograph. The choice of the objective function for model calibration directly influences which trade-offs are made. The calibration of a rainfall-runoff model using the Nash-Sutcliffe efficiency is known to give higher importance to fitting flow peaks because of its quadratic formulation, and this is reflected in the generally

better performance of a model calibrated on such a criterion to predict streamflow characteristics for high flow conditions (Shrestha et al., 2014). Composite objective functions such as the Kling-Gupta efficiency (KGE) are now often preferred, since KGE explicitly considers linear correlation, bias, and variability in a balanced or customisable way (Gupta et al., 2009; Kling et al., 2012). Nonetheless, the quadratic formulation on flows remains in the linear correlation component of KGE, and a prior transformation of flow values is often suggested, for example to put more emphasis on low flows (Santos et al., 2018). In

order to improve the model calibration for the predictions of different ranges of flows, the whole or parts of the flow duration curve have also been found useful in improving the model simulation of the whole hydrograph (e.g. Westerberg et al., 2011; Pfannerstill et al., 2014). However, the flow duration curve does not embed any information about the timing or duration of flow events, which can be essential for ecological species (Arthington et al., 2006).

In order to improve the prediction of a diverse range of SFCs (e.g. both high flow and low flow conditions, or both magnitude,

and timing/duration of flows), multi-objective calibration methods applied to flows (referred to as traditional objective functions

hereafter) have already been explored. Vis et al. (2015) found that certain combinations of objective functions fitted to flows, each focussing on different statistical aspects of the streamflow hydrograph (e.g. volume error, correlation), tend to be more suitable for the prediction of the magnitude of average flows, and the timing of moderate flows and low flows than using a single objective function fitted to flows, e.g. NSE alone. But the authors found that, on average, NSE calibration produces
the smallest errors on 12 SFCs, and they did not find a single best calibration strategy fitted to flows for predicting all SFCs at once. Garcia et al. (2017) identified that an average sum of KGE and KGE on inverted flows has better skill than KGE alone, or KGE of inverted flows alone to predict seven SFCs relative to low flow conditions. Hernandez-Suarez et al. (2018) found that a three-objective-function calibration strategy with NSE, NSE of the square root transformed flows, and NSE on the relative deviations is capable of predicting 128 SFCs within a $\pm$ 30 % error range in their study catchment, with larger errors
on SFCs for extreme high and low flows conditions. Mizukami et al. (2019) compared objective functions to predict the bias in annual peak flows, and they found that KGE was better suited than NSE because it is better at reproducing the flow variability, reducing the underestimation of high flows, while not fully eliminating it. These studies suggest that combinations of traditional objective functions (e.g. NSE, KGE) on transformed and untransformed streamflow series can improve the prediction of SFCs compared to single objective calibration strategies, while the predictions of extreme flow conditions remain problematic.

To further improve the prediction of a range of SFCs, a pragmatic approach is to directly use an objective function fitted to the target SFCs (referred to as tailored objective functions hereafter) in the expectation that this will improve predictions of these same SFCs. Mizukami et al. (2019) found that the annual peak flow bias was best predicted by using the bias itself as the objective function (to be minimised), outperforming KGE, and other KGE formulations with more weight on its flow variability component, but they also found that using this single SFC as an objective function resulted in overfitting, reducing its
transferability in time. Pool et al. (2017) also found that a given SFC is the best objective function to predict itself, and this for 13 different SFCs, outperforming NSE. The authors also used a four-SFC metric as the objective function, but they found that the prediction of other SFCs not included in this objective function was not improved compared to NSE. Kiesel et al. (2017) used a seven-SFC metric as objective function to predict these seven SFCs, and found that this objective function outperformed KGE on almost all seven SFCs. The authors also found that for two of these seven SFCs used as a single objective function
produced better overall performance than KGE too. Zhang et al. (2016) found that a 16-SFC metric used as the objective function outperformed the RMSE used as a single objective function, especially for the prediction of SFCs on low flow and high flow events. On the other hand, Garcia et al. (2017) reached a different conclusion to predict seven SFCs focussing on low flow conditions, where their combined seven-SFC metric was not as robust as the composite objective function made of KGE and KGE on inverted flows. However, unlike the other studies previously mentioned, this multi-SFC metric focussed
solely on low flow conditions, which could explain its lack of robustness, given the difficulty in predicting extreme streamflow conditions well.

    Hydrological models are generally found to be less accurate than regional regression models in predicting particular SFCs because separate regression models can be purposely developed for each target SFC individually (Murphy et al., 2013). Similar behaviour has been found for calibrated rainfall-runoff models, where specific calibration focussed on the target SFC is the
best performing calibration option for that SFC (Kiesel et al., 2017; Pool et al., 2017; Mizukami et al., 2019). However, when

calibrating on a specific SFC, while the model's ability to predict that indicator may improve, the physical representation of the catchment's overall behaviour, captured in the effective parameter values of the model, could be compromised, preventing the use of the model for predicting other indicators. For instance, Pool et al. (2017) found that using a combination of SFCs as an objective function does not perform as well as the Nash-Sutcliffe efficiency fitted to flows to predict SFCs not included in the combination, and the authors suggested that the use of SFCs in calibration may not produce consistent model parameter sets. Poff and Zimmerman (2010) and Knight et al. (2014) showed that each stream species is sensitive to its own combination of SFCs relating to its own preferences for living conditions, which constitutes the ecological flow regime (Knight et al., 2012). When several species are considered simultaneously, the number of SFCs to predict is likely to increase, even though some stream species may respond to broadly similar streamflow characteristics. If the number of SFCs to predict were to increase, it could be expected that using traditional objective functions would be a more parsimonious calibration strategy and that more targetted characteristics of the hydrograph would be predicted well. For example, Archfield et al. (2014) found that a set of seven streamflow statistics characterising the stochastic properties of daily streamflow series, like traditional objective functions fitted to flows, is more parsimonious than a set of 33 SFCs to classify stream gauges for hydro-ecological purposes.

The objectives of this study are to assess whether tailored objective functions are the best calibration strategy to predict the SFCs they contain by targetting three vectors of SFCs of varying size for the first time on a common set of 33 Irish catchments, and on 14 different calibration-evaluation periods. Moreover, a fourteen split-sample test is applied to allow for more extensive comparison of the skills of the objective functions for model calibration. Also, a four-fold analysis of model overall performance, model performance stability, model robustness, and model consistency is undertaken to assess the parsimonious character of tailored objective functions. In additional, the skill of the objective functions are compared on a set of 156 SFCs and on nine percentiles of the flow duration curve to extend the comparison beyond the SFCs contained in the tailored objective functions and explore trends on specific categories of streamflow characteristics. The conceptual SMART rainfall-runoff model (Mockler et al., 2016) is used to simulate streamflow in the 33 study catchments and three traditional objective functions fitted to flows are compared with three tailored objective functions fitted to SFCs.

## 2 Data and model

### 2.1 Streamflow characteristics

The selection of streamflow characteristics used in the tailored objective functions relies on previous studies that identified sets of SFCs representative of the habitat preferences of fish communities in the Southeastern US (Knight et al., 2014; Pool et al., 2017), and of invertebrate communities in Germany (Kakouei et al., 2017; Kiesel et al., 2017). In addition, a third set of SFCs is formed from the union of the two first sets of SFCs, assuming that invertebrate and fish communities are sensitive to two mainly distinct habitat preferences, so that a larger set of SFCs would need to be considered in this case. These three sets are assumed to be of some ecological relevance to the Irish study catchments for the purpose of comparing traditional and plausible tailored objective functions, however, direct empirical evidence is lacking to date to confirm whether they are the optimal indices for key invertebrate and fish species found in Irish rivers.

The indices are listed and detailed in Table 1. Except for q85 that is directly derived from the flow duration curve, all streamflow characteristics are defined in Olden and Poff (2003) and their calculation follows the method implemented in the R-package EflowStats (Henriksen et al., 2006; Archfield et al., 2014). However, all computations for this study were carried out in Python where the NumPy package was used to vectorise the calculations of the SFCs (i.e. to formulate the calculations as arithmetic operations between vectors and matrices). This makes use of algorithms directly coded in C, avoiding the redundant interpretation of Python or R statements in loops which is significant for iterations over a very large number of streamflow time series (i.e. $8.316 \times 10^6$) (Hallouin, 2019a).

## 2.2 Study catchments

This study used discharge records with a minimum of 14 hydrological years with complete daily discharge data in the period from the 1[st] of October 1986 to the 30[th] of September 2016. If any daily value was missing, the hydrological year was discarded. The calculation of some streamflow characteristics requires a strictly continuous daily streamflow time series and it can be difficult to find time series with no missing discharge measurement at all. The length of 14 years was set as the minimum requirement in order to have seven years for calibration and seven years for evaluation for each catchment. A minimum calibration period length of five years is recommended by Merz et al. (2009) to capture the temporal hydrological variability.

The data availability for the gauges meeting these requirements is presented on Figure A1. In most catchments, these 14 hydrological years were not necessarily consecutive. For catchments featuring more than 14 complete hydrological years, the additional available years were not used in order to avoid any positive or negative bias due to differences in data series length between these catchments and other catchments in the set. The daily discharge data used in this study is provided by the Office of Public Works (2019), and Ireland's Environmental Protection Agency (2019).

Catchment selection was also influenced by the quality of the discharge data, including the quality of the rating curve at the gauge as determined by Webster et al. (2017). Heavily regulated rivers were discarded. A total of 33 gauges (displayed on Figure 1b) featured sufficient data of good quality to be used as study catchments, amongst which 18 are nested within any of the 15 distinct catchments. The 15 distinct catchments (displayed on Figure 1a) cover 26 % of the Republic of Ireland. They are located throughout the country, and they represent a diversity of Irish soils and geology (Figure 1c,d), despite lacking some of the most elevated catchments in the Wicklow mountains (relief on the East coast), and the mountainous edge on the Atlantic coast (Figure 1b). Their average annual rainfall ranges from 916 to 1660 $\mathrm{mm\,yr^{-1}}$, and the average annual potential evapotranspiration varies from 497 to 578 $\mathrm{mm\,yr^{-1}}$. The size of the catchments varies from 25 to 2462 $\mathrm{km^2}$, while their average elevation ranges from 5 to 910 m above sea level, and their average slope ranges from 19 to 121 $\mathrm{m\,km^{-1}}$. Estimated baseflow indices range from 0.31 to 0.79 (see Table A1 for full details).

## 2.3 Rainfall-runoff model

The Soil Moisture Accounting and Routing for Transport (SMART) model used here is an enhancement of the SMARG lumped, conceptual, rainfall-runoff model (Soil Moisture Accounting and Routing with Groundwater) developed in University College Galway (Kachroo, 1992) and based on the soil layers concept (O'Connell et al., 1970; Nash and Sutcliffe, 1970).

Separate soil layers were introduced to capture the decline with soil depth in ability of plant roots to extract water for evapotranspiration. SMARG was originally developed for flow modelling and forecasting and was incorporated into the Galway Real-Time River Flow Forecasting System (GFFS) (Goswami et al., 2005). The SMART model reorganised and extended SMARG to provide a basis for water quality modelling by separating explicitly the important flow pathways in a catchment, needed for an EPA funded project "Pathways", and it has been successfully fitted to over 30 % of Irish catchments (Mockler et al., 2016).

The routing component distinguishes between five runoff pathways: overland flow, drain flow, interflow, shallow groundwater flow, and deep groundwater flow (Figure 2). It usually runs at an hourly or daily time-step, requires inputs of measures of precipitation and estimates of potential evapotranspiration, and produces estimates of discharge from the catchment. It normally has ten free parameters (Table 2). During energy-limited periods (i.e. when potential evapotranspiration is less than incident rainfall), the model first estimates effective or excess rainfall by applying a scaling correction $\theta_T$ and subtracting any direct evaporation. A threshold parameter $\theta_H$ determines how much (if any) of this becomes direct surface runoff through the Horton (infiltration excess) mechanism. Any surplus rainfall is assumed to infiltrate the top layer of the soil. The soil is modelled as six layers with a total soil moisture capacity of $\theta_Z$. As the capacity of a layer is exceeded, surplus moisture moves to a deeper layer if it has capacity or else is intercepted by drains or moves to the shallow or deep groundwater stores. In water-limited periods (i.e. when potential evapotranspiration exceeds incident rainfall), the model attempts to meet the evapotranspiration demand by supplying moisture from the soil layers, starting from the top layer but when this is dry from lower layers but with an increasing difficulty expressed by a parameter $\theta_C$. Each of the above pathways is modelled as a single linear reservoir, each with its own parameter ($\theta_{SK}$ for overland and drain flow, $\theta_{FK}$ for interflow, $\theta_{GK}$ for shallow and deep groundwater flow). The outputs of all of these are routed through a single linear reservoir representing river routing ($\theta_{RK}$). Note, a detailed description of the conceptual model is provided in the Supplement.

## 3 Method

### 3.1 Split-sample tests

Split-sample tests are commonly used to analyse the performance of hydrological models (Klemeš, 1986). Coron et al. (2012) proposed a generalised split-sample test using a sliding window for calibration, and evaluating the model performance on all other independent windows in the period. de Lavenne et al. (2016) adapted this strategy to calibrate a catchment model with a sliding window, and to evaluate the simulations on all other years (before and/or after the sliding window) in the study period. These approaches have the advantage of reducing any influence of different calibration/evaluation periods, compared with a single split-sample test that divides the study period into fixed, separate, calibration and evaluation periods.

The split-sampling strategy in this study is adapted from the original approach by de Lavenne et al. (2016) in that it uses each hydrological year the same number of times in each of the 14 split-sample tests. For each catchment, the 14-hydrological-year series of discharge measurements is split into two seven-hydrological-year periods, and the split is repeated 14 times (Figure 3). It is implicitly assumed that any combination of hydrological years can be used, even if they are not consecutive. Thus, there

are theoretically 3432 different combinations of seven-year periods in a 14-year study period. These combinations would be expected to represent all possible climatic combinations represented in the data for the study period; however, given the large dataset this would generate, it was decided to work only on 14 combinations by using the window of seven consecutive years, rather than more sophisticated boot-strapping strategies.

## 3.2 Model setup

The SMART model is forced with daily rainfall and daily potential evapotranspiration provided by the national meteorological office Met Éireann (2019). The potential evapotranspiration is calculated by Met Éireann using the FAO Penman-Monteith formula (Allen et al., 1998) with coefficients adjusted for Irish conditions and meteorological data at their synoptic weather stations. A five-hydrological-year warm-up period is used to determine the initial states of the soil layers and reservoirs in the model. The five-year warm-up period is applied prior the first complete hydrological year used in the split-sample test on Figure A1. A Python implementation of the SMART model (Hallouin et al., 2019) is used to simulate the hydrological response in all study catchments from the first day of the first warm-up year until the last day of the fourteenth complete hydrological year. The corresponding calibration and evaluation periods are then extracted from these time series as required (see Figure 3).

## 3.3 Model calibration

The calibration of the model is done using six different objective functions. The calibration procedure is illustrated in Figure 4, steps (a) to (d). This methodology is applied for each study catchment individually. First, in step (a), the model parameter space is explored using a Latin Hypercube Sampling (LHS) strategy (McKay et al., 1979) to generate $10^5$ random parameter sets well distributed in the parameter space. The feasible parameter ranges used to define the boundaries of the parameter space are based on a previous study by Mockler et al. (2016) providing typical ranges for Irish catchments. The model is then used in step (b) to simulate the catchment response with each of these $10^5$ parameter sets, which produces as many hydrographs.

In step (c) (Figure 4), six different objective functions are used to calculate the model performance by comparing the simulated and observed catchment responses. Three variants of the Kling-Gupta efficiency (Gupta et al., 2009) are tested. First, the KGE criterion is computed on the untransformed discharge series, that is $E_{KG}^{Q}$ (Equation 1); since the linear correlation coefficient included in KGE is more sensitive to errors on flow peaks (Krause et al., 2005), it is considered to put more emphasis on high flow conditions. Second, the KGE criterion is computed on the inverted discharge series, that is $E_{KG}^{Q^{-1}}$ (Equation 2); this objective function on transformed flows is used to put more emphasis on low flow conditions. Inverted flows are preferred over log-transformed flows in order to retain the dimensionless character of the objective function allowing for comparison across catchments, however, any transformation of flows before computing KGE leads to the loss of the physical interpretability of the three components of KGE (Santos et al., 2018), but it is not required here. Third, the KGE criterion is computed from the square root of the discharge series, that is $E_{KG}^{Q^{0.5}}$ (Equation 3); this objective function is used in order to put more emphasis on moderate flow conditions, by reducing the weighting of low flow or high flow conditions (Garcia et al., 2017). These variants of KGE, referred to as traditional objective functions hereafter, assess the suitability of the model parameters by comparing the entirety of the observed and simulated hydrographs. In addition, three combinations (vectors) of streamflow characteristics (SFCs)

are used as tailored objective functions, and referred as such hereafter. They assess the model performance by comparing the observed and simulated values of the SFCs extracted from the observed and simulated hydrographs, respectively. For each vector of SFCs, the Euclidean distance (Equation 4) separating the observed and simulated points in the multi-dimensional space formed by each dimension in the vector of SFCs is calculated, and this distance is subtracted from one so the efficiency measure has an optimum at one, like KGE. One efficiency is calculated from the set of SFCs identified by Kiesel et al. (2017) $E_{\mathrm{SFC}}^{\mathrm{K}}$ (vector of 7 SFCs as target), one is calculated from the set identified by Pool et al. (2017) $E_{\mathrm{SFC}}^{\mathrm{P}}$ (vector of 13 SFCs as target), and one is calculated from both sets of SFCs at once $E_{\mathrm{SFC}}^{\mathrm{KP}}$ (vector of 18 SFCs as target). Similar to Kiesel et al. (2017), each SFC is normalised prior the calculation of the Euclidean distance (Equations 5, 6) so that its value is bounded between 0 and 1, effectively giving all SFCs the same weight in the computation of the Euclidean distance. Each of these six objectives functions are used to produce $10^5$ efficiency scores for all the calibration cases.

Eventually, in step (d), the best 1% parameter sets (i.e. those with the highest efficiency scores) are retained as "behavioural" on the basis of their performance on the chosen objective function, which yields a set of $10^3$ parameter sets. This calibration approach is similar to the GLUE methodology (Beven and Binley, 1992, 2014) without a threshold for acceptability to characterise the behavioural character of a parameter set. Instead of defining a threshold of acceptability, as in the GLUE method, here it is preferred to analyse the statistics of equally sized parameter sets (i.e. 1 % best) with each of the different objective functions.

In order to give some perspective on the absolute performance of each of the six objective functions beyond the relative comparison, a benchmark is defined by randomly sampling $10^3$ parameter sets in the generated Latin Hypercube. This benchmark corresponds to an uninformative calibration, and will be referred to as R in the Results section. This follows the recommendations made by Seibert et al. (2018) to define a lower benchmark when assessing the performance of a hydrological model, because any model should be expected to reproduce some of the streamflow variability simply due to the use of observed forcing data specific to the catchment of interest. If the performance of the calibrated model does not exceed the performance of the benchmark, then the suitability of the model and/or its calibration is questionable.

$$
\begin{aligned}
E_{KG}^{Q} &= E_{KG}\big(q_{obs}, q_{sim}\big) \\
&= 1 - \sqrt{(r-1)^2 + (\alpha-1)^2 + (\beta-1)^2} \\
&= 1 - \sqrt{\left(\frac{\mathrm{cov}(q_{obs}, q_{sim})}{\sigma_{q_{obs}} \cdot \sigma_{q_{sim}}} - 1\right)^2 + \left(\frac{\sigma_{q_{sim}}}{\sigma_{q_{obs}}} - 1\right)^2 + \left(\frac{\mu_{q_{sim}}}{\mu_{q_{obs}}} - 1\right)^2}
\end{aligned}
\tag{1}
$$

$$
E_{KG}^{Q^{-1}} = E_{KG}\left(\frac{1}{q_{obs} + 0.01 \cdot \mu_{q_{obs}}}, \frac{1}{q_{sim} + 0.01 \cdot \mu_{q_{sim}}}\right)
\tag{2}
$$

$$
E_{KG}^{Q^{0.5}} = E_{KG}\big(\sqrt{q_{obs}}, \sqrt{q_{sim}}\big)
\tag{3}
$$

where cov, $\sigma$, and μ correspond to the covariance, the standard deviation, and the arithmetic mean, respectively; $q_{obs}$, and $q_{sim}$ correspond to the time series of observed discharge, and simulated discharge, respectively. Noteworthy, a constant is added to the inverted discharge values in Equation 2 in order to avoid zero flows issues, and a hundredth of the arithmetic mean of the corresponding discharge series is used as recommended by Pushpalatha et al. (2012).

$$E_{SFC}^{target} = 1 - \sqrt{\sum_{j=1}^{N_{target}} \left( c_{obs,j}^* - c_{sim,j}^* \right)^2} \tag{4}$$

where $N_{target}$ corresponds to the number of SFCs contained in the targetted combination of SFCs (the specific SFCs contained in each targetted combination can be found in Table 1), and where $c_{obs,j}^*$, $c_{sim,j}^*$ correspond to the $j^{th}$ observed SFC value in the combination, and the $j^{th}$ simulated SFC value in the combination, respectively, which were normalised as described in Equation 5, and in Equation 6, respectively.

$$c_{obs,j}^* = \frac{c_{obs,j} - \min\left(c_{obs,j}; \left\{c_{sim_i,j}\right\}_{i=1}^{10^5}\right)}{\max\left(c_{obs,j}; \left\{c_{sim_i,j}\right\}_{i=1}^{10^5}\right) - \min\left(c_{obs,j}; \left\{c_{sim_i,j}\right\}_{i=1}^{10^5}\right)} \tag{5}$$

$$c_{sim_i,j}^* = \frac{c_{sim_i,j} - \min\left(c_{obs,j}; \left\{c_{sim_i,j}\right\}_{i=1}^{10^5}\right)}{\max\left(c_{obs,j}; \left\{c_{sim_i,j}\right\}_{i=1}^{10^5}\right) - \min\left(c_{obs,j}; \left\{c_{sim_i,j}\right\}_{i=1}^{10^5}\right)} \tag{6}$$

where $c_{obs,j}$, $c_{sim_i,j}$ correspond to the $j^{th}$ observed SFC value in the combination, and the $j^{th}$ simulated SFC value in the combination for the $i^{th}$ streamflow simulation amongst the Latin Hypercube sample, respectively.

## 3.4  Model evaluation

The method used to evaluate the performance of the predictions with a model calibrated with each of the six different objective functions is described in steps (e) to (h) of Figure 4. Again, this methodology is applied for each study catchment individually. First, in step (e), the model is run separately with each of the behavioural ($10^3$) model parameter sets to simulate its catchment response in the evaluation period, which produces $10^3$ hydrographs. From each hydrograph, in step (f), the performance of the model prediction in evaluation is assessed with any of the six objective function described in subsection 3.3, which yields $10^3$ efficiency scores on the evaluation period. Finally, in order to compare the predictive performance in each catchment, in step (g) a measure of central tendency, the median, is used to summarise the performance of the behavioural parameter sets identified with each of the six objective functions. From there, different analyses are carried out to explore the comparative skills of the six objective functions considered, and they are detailed below.

### 3.4.1  Overall performance

First, the overall performance in evaluation of the model calibrated with each of the six objective functions is assessed by averaging across the 14 split-sample tests the median efficiency scores obtained in step (g) (Figure 4), and then averaging

again across the 33 study catchments in order to compare the calibration skills of the various objective functions. Since all six objective functions are defined as Euclidean distances subtracted from one, the overall performance ranges from $-\infty$ to one, with an optimal value at one.

The skills of the six objective functions to calibrate the model are first compared using the traditional objective functions as efficiency scores for the evaluation period in subsection 4.1 in order to assess whether they are capable of reproducing the shape and timing, the variability, and the average volume of the observed hydrograph. Because a hydrological model is used to make the streamflow predictions, it is important to check whether the different objective functions, especially the tailored ones, are capable of finding parameter values that are able to reproduce the catchment hydrological response relatively well. Moreover, this gives more confidence in the model structure as being a plausible approximation of the relevant hydrological processes in the study catchments.

The skills of the six objective functions to calibrate the model are then compared using the tailored objective functions as efficiency scores for the evaluation period in subsection 4.2 in order to assess their suitability to be used as calibration targets for the prediction of sets of streamflow characteristics, which is the primary focus of this study.

### 3.4.2 Performance stability

The use of fourteen split-sample tests allows for the assessment of the model performance on evaluation periods that are different (either completely different or at least partially, see Figure 3). As a result, the stability of the performance in evaluation can be explored, which is important to have confidence that the model performance is independent of the study period.

The stability is calculated from the standard deviation of the median efficiency scores across the 14 split-sample tests, and is then averaged across the 33 study catchments in subsection 4.3 to obtain a measure of the overall stability of the model performance. This is done for all the models calibrated with each of the six objective functions. The stability ranges from zero to $+\infty$, with an optimal value at zero.

### 3.4.3 Performance robustness

Additionally, the objective functions are compared in relation to their ability to retain their fitting skill found in calibration, where observed data is available to the optimising process, with the performance in the evaluation period, in order to assess the temporal robustness of the model performance. Robustness analysis is important to check for model overfitting to the calibration data, which could reduce the predictive power of the model.

The robustness is calculated from the difference between the median efficiency in calibration and the median efficiency in evaluation, then averaging the difference across the 14 split-sample tests, and finally averaging across the 33 study catchments in subsection 4.4 to obtain the robustness with each of the six objective functions. The stability ranges from $-\infty$ to $+\infty$, with an optimal value at zero. Stability is typically expected to be positive because the performance in calibration, where observed data is used, is expected to exceed the performance in evaluation, where the model is unaware of the observed data.

### 3.4.4 Model consistency

Finally, the model consistency obtained with each of the six objective functions is explored. The concept of consistency has been previously used as a guide in the selection from competing model structures (Euser et al., 2013). Originally used as the capacity of a model structure to predict a range of hydrological signatures with the same parameter set, the idea of consistency is applied to the objective functions in this study. It is used to compare different objective functions according to their ability to identify the same parameter sets as behavioural across the 14 split-sample tests, described above. Consistency establishes whether similar performance results were obtained with largely different parameter sets.

The consistency is calculated from the ratio of the number of model parameter sets identified as behavioural that are common to all 14 split-sample tests divided by the total number of behavioural parameter sets (i.e. $10^3$), and then the average ratio across the 33 study catchments is calculated in subsection 4.5 to obtain the model consistency with a given objective function used to identify the behavioural parameter sets. The consistency ranges from zero to one, with an optimal value at 1.

It is to be noted that, in order to be able to assess the model consistency, a single Latin Hypercube sampling of the $10^5$ parameter sets per catchment is used. That is to say that the Latin Hypercube is generated once and it is used on the 14 different calibration-evaluation periods in order to be able to determine whether a behavioural parameter set identified as behavioural on one test remains behavioural on a different test.

### 3.4.5 Analysis on the components of the objective functions

In order to investigate the reasons for the trends identified in model performance, stability, robustness, and consistency, an analysis of the performance of the six objective functions to predict the shape and timing, the variability, and the bias of the observed hydrograph is carried out by using the three components, r, $\alpha$, and $\beta$ of $\mathrm{E_{KG}^Q}$, respectively Equation 1. Because of the transformation applied to the discharge series in $\mathrm{E_{KG}^{Q^{0.5}}}$ and $\mathrm{E_{KG}^{Q^{-1}}}$, the direct physical interpretation of their three components is lost (Santos et al., 2018), so they are not analysed further.

In addition, an analysis of the performance of the six objective functions to predict each individual SFC is carried out by calculating the absolute normalised error between the simulated and the observed SFC values (Equation 7).

$$
e^*_{sim_i,j} = \left| \frac{c_{sim_i,j} - c_{obs,j}}{\max\left(c_{obs,j}; \{c_{sim_i,j}\}_{i=1}^{10^5}\right) - \min\left(c_{obs,j}; \{c_{sim_i,j}\}_{i=1}^{10^5}\right)} \right| \tag{7}
$$

where $c_{obs,j}$, $c_{sim_i,j}$ correspond to the $j^{th}$ observed SFC value in the combination, and the $j^{th}$ simulated SFC value in the combination for the $i^{th}$ streamflow simulation amongst the Latin Hypercube sample, respectively.

For each component analysed, the same approach as the one used for assessing the overall model performance in subsubsection 3.4.1 is chosen, i.e. the median value of a given component for the behavioural parameter set is calculated, it is then averaged across the 14 split-sample tests, and it is finally averaged across the 33 study catchments to obtain an overall skill of each objective function in predicting these individual components.

## 4 Results

### 4.1 Are the candidate objective functions capable of reproducing the catchment hydrograph?

The SMART model calibrated on $E_{KG}^Q$ is found to be able to reproduce the observed catchment hydrographs reasonably well in all 33 study catchments, with average $E_{KG}^Q$ scores in calibration across the 14 split-sample tests ranging from 0.58 to 0.94
with a median of 0.86.

On average, all six objective functions perform well in reproducing the observed hydrograph when more weight is given to predicting high flows well, with $E_{KG}^Q$ scores in evaluation between 0.69 and 0.82 in evaluation (Figure 5a). They largely outperform the average benchmark score of 0.40, indicating that all six objective functions are useful to find parameter sets representative of the hydrological behaviour of our catchments. Using $E_{KG}^Q$ for calibration is found to be the best objective
function when measured using $E_{KG}^Q$ with a score of 0.82, followed by $E_{KG}^{Q^{0.5}}$ with a score of 0.80. However, $E_{KG}^{Q^{-1}}$ is outperformed by any of the three tailored objective functions. $E_{SFC}^{KP}$ is the best tailored objective function when measured on $E_{KG}^Q$, followed by $E_{SFC}^P$, and $E_{SFC}^K$. This can be explained by the fact that $E_{SFC}^{KP}$ and $E_{SFC}^P$ contain a majority of SFCs for high flow conditions (Table 1), while $E_{SFC}^K$ contains a majority of SFCs for low flow conditions.

When more importance is given to predicting average flow conditions, i.e. using $E_{KG}^{Q^{0.5}}$ (Figure 5b), this is $E_{KG}^{Q^{0.5}}$ that is the
best performing objective function with 0.87, followed by $E_{KG}^Q$ with 0.86, and the three tailored objective functions with very comparable performances (between 0.84 and 0.85). Since the three tailored objective functions contain comparable proportions of SFCs for average flow conditions (i.e. around 30%, see Table 1), it can explain why their values of $E_{KG}^{Q^{0.5}}$ are close.

While $E_{KG}^{Q^{-1}}$ is the worst objective function when assessed on $E_{KG}^Q$ or $E_{KG}^{Q^{-1}}$, when more emphasis is put on low flows, i.e. assessed using $E_{KG}^{Q^{-1}}$ (Figure 5c), it performs the best out of the six objective functions with a score of 0.67, followed by $E_{SFC}^K$
with a score of 0.59. $E_{KG}^{Q^{0.5}}$ and $E_{SFC}^K$ perform similarly with a score of 0.56, while $E_{SFC}^P$ has a score of 0.55. $E_{KG}^Q$ is the worst objective function to choose out of the six to predict the hydrograph with more emphasis on predicting low flows well. Nonetheless, it remains largely better than the lower benchmark and its score at 0.07. Again, the proportion of SFCs for low flow conditions can explain the ranking of the three tailored objective functions, where $E_{SFC}^K$ features a majority of SFCs for low flow conditions (3 out of 7 SFCs, see Table 1), while $E_{SFC}^P$ features the lowest proportion of SFCs for low flow conditions
(i.e. 4 out of 13, Table 1).

### 4.2 Which objective function provides the most accurate SFC predictions?

The comparison reveals that the differences in performance between most objective functions are small (Figure 6a-c). However, their performances are largely exceeding those of the benchmark on all three tailored objective functions, indicating that all six objective functions are similarly informative for model calibration. The best predictive performance in evaluation for a
given set of SFCs targetted in this study is always obtained using this same combination of SFCs as the objective function in calibration, e.g. the best $E_{SFC}^K$ score in evaluation (0.74) is obtained with $E_{SFC}^K$ as the objective function for calibration. ($E_{SFC}^P$ scores 0.56 on $E_{SFC}^P$, $E_{SFC}^{KP}$ scores 0.50 on $E_{SFC}^{KP}$). Furthermore, the largest combination featuring 18 SFCs $E_{SFC}^{KP}$ is a

competitive option, even when the focus is on smaller subsets of SFCs (i.e. scores 0.73 on $E_{SFC}^{K}$, or scores 0.56 on $E_{SFC}^{P}$), and it outperforms any of the three formulation of $E_{KG}$.

The best performing traditional objective function to predict any of the three sets of SFCs is consistently found to be $E_{KG}^{Q^{0.5}}$, with scores of 0.72 on $E_{SFC}^{K}$, 0.54 on $E_{SFC}^{P}$, and 0.48 on $E_{SFC}^{KP}$. On the other hand, $E_{KG}^{Q^{-1}}$ is found to be the worst performing traditional objective function, with scores of 0.67 on $E_{SFC}^{K}$, 0.41 on $E_{SFC}^{P}$, and 0.34 on $E_{SFC}^{KP}$. Given that $E_{SFC}^{K}$ contains a majority of SFCs for low flow conditions (3 out of 7), this is surprising to find the traditional function with the strongest focus on predicting low flow conditions is the worst performing one. However, Garcia et al. (2017) also found that $E_{KG}^{Q^{-1}}$ is not the best to predict low-flow indices, and the authors recommend an arithmetic mean of $E_{KG}^{Q}$ and $E_{KG}^{Q^{-1}}$ as a better alternative to predict them.

In addition, the dispersion of the performance across the 33 study catchments, measured by standard deviation (represented as error bars on Figure 6a-c), is smaller for the better performing objective functions, which indicates that in addition to predict well on average, they produce less variability in performance across the different study catchments.

### 4.3 Which objective function provides the most stable SFC predictions?

The average stability of the performance across the 14 split-sample tests also shows only small differences between the different objective functions (Figure 6d-f). Moreover, the absolute stability scores, measured by the standard deviation across the 14 split-samples (see subsubsection 3.4.2), are also relatively small, i.e. never exceeding 0.05. It is also important to note that the benchmark shows comparable performance stability with the six objective functions used for calibration. This suggests that stability is not very useful here to compare the objective functions, given that an uninformative calibration yields similar stability. This also implies that the differences observed in terms of overall performance are not dependent on the study period considered, since stability values are small.

### 4.4 Which objective function provides the most robust SFC predictions?

The analysis of the robustness of the different objective functions to predict each of the three sets of SFCs uncovers a general trend whereby traditional objective functions are more robust than tailored objective functions, i.e. the drop in performance from the calibration period to the evaluation period is smaller for the traditional objective functions (Figure 6g-i).

The average drop in performance is consistently below 0.01 for $E_{KG}^{Q}$, $E_{KG}^{Q^{0.5}}$, and $E_{KG}^{Q^{-1}}$, on all three tailored objective functions used as efficiency scores in evaluation. On the other hand, the largest drop in performance on any of the three tailored objective functions used as evaluation targets is always obtained with this same tailored objective function used in calibration. For instance, $E_{SFC}^{K}$ shows an average drop of 0.045 on $E_{SFC}^{K}$, while the drop is only 0.016 with $E_{SFC}^{P}$, and 0.022 with $E_{SFC}^{KP}$. These results suggest that the tailored objective functions suffer from more overfitting issues in calibration that the traditional objective functions to predict the three set of SFCs. Nonetheless, the tailored objective functions remain the best performing options when considering results in the evaluation period, so that although they reach better fitting in calibration but at the cost of larger performance drops from calibration to evaluation. These results are consistent with Garcia et al. (2017) who found that their tailored objective function made of 7 SFCs was not robust either.

## 4.5 Which objective function yields the most consistent behavioural parameter sets?

Unlike the measures of average model performance and performance stability, the consistency measures reveal more significant differences between the six objective functions compared here (Figure 7). On average, $E_{KG}^{Q^{0.5}}$ and $E_{KG}^{Q}$ clearly outperform all other objective functions with a consistency exceeding 0.5 (0.52 and 0.51, respectively). This means that more than half of the behavioural parameter sets identified with these two objective functions remain selected as behavioural across all 14 split-sample tests. $E_{KG}^{Q^{-1}}$ comes second last with a consistency of 0.19. The two objective functions yielding the lowest consistencies are $E_{KG}^{Q^{-1}}$ and $E_{SFC}^{K}$, with 0.19 and 0.13, respectively. The main similarity between these two objective functions is that they focus more on low flow conditions than average or high flow conditions. This may be the reason for their lack of consistency.

The consistency ratios for the tailored objective functions appear to be related to the number of SFCs they contain. Indeed, $E_{SFC}^{K}$ containing only seven SFCs comes last with a consistency of 0.13, $E_{SFC}^{P}$ with 13 SFCs has a consistency of 0.31, and $E_{SFC}^{KP}$ containing all 18 SFCs has a consistency of 0.34. However, given that only three set of SFCs are tested, this could only be a coincidence, and additional research on the impact of the number of components contained in the tailored objective functions on the model consistency is required.

## 4.6 Are there specific components of the objective functions limiting their performances?

### 4.6.1 Shape and timing, variability, bias

First, comparing the six objective functions on the three components of $E_{KG}^{Q}$ (Figure 8) reveals that the shape and timing (r) is the most difficult aspect of the hydrograph to predict, while the total volume ($\beta$) is the least difficult. The flow variability ($\alpha$) is consistently underestimated, while the total volume is overestimated with all but one objective function (i.e. $E_{KG}^{Q^{-1}}$).

$E_{KG}^{Q^{-1}}$ performs the worst on two of the three components, with a score of 0.836 on the linear correlation r, and 0.882 on the variability $\alpha$ which indicates that it is the objective function struggling the most to reproduce the shape and timing of the observed hydrograph, and also the one that most underestimates the observed spread of flows. On the other hand, it is the best objective function to use in calibration to estimate the volume of water at the catchment outlet (score of 0.997 on $\beta$) and, in fact, the only one to underestimate this volume. On the other hand, $E_{KG}^{Q}$ and $E_{KG}^{Q^{0.5}}$ perform well on $\alpha$ and $\beta$, while it is not as good on the correlation coefficient r. Nevertheless, they are better than most other objective functions on r, with amongst the highest scores on r (0.894 and 0.888, respectively).

The low performance of $E_{SFC}^{K}$ found in subsection 4.1 can be mainly attributed to a lower correlation component of $E_{KG}^{Q}$, with a value of 0.863, and to a lesser extent to failure in capturing the flow variability ($\alpha$ value of 0.927). Even though $E_{SFC}^{K}$ is the worst objective function for the bias ($\beta$ value of 1.038), this is its best component of $E_{KG}^{Q}$. $E_{SFC}^{P}$ and $E_{SFC}^{KP}$ show strong skills in capturing the flow variability, with $\alpha$ values close to one (0.953 and 0.951, respectively), they show comparable skills as $E_{KG}^{Q}$ and $E_{KG}^{Q^{0.5}}$ on the correlation coefficient, while they overestimate the total volume the most (1.018 and 1.022 for the bias, respectively).

$E_{SFC}^{K}$ and $E_{KG}^{Q^{-1}}$ share in common a stronger weight for low flow periods compared with moderate and high flow periods, which is likely the reason compromising their performance on the linear correlation coefficient r which is known to give more

weight to high flow periods, that typically exhibit larger errors, that are amplified by the quadratic formulation of the correlation coefficient (Krause et al., 2005). Moreover, $E_{SFC}^P$ and $E_{SFC}^{KP}$ contain higher proportions (and larger numbers) of SFCs relating to flow magnitude than $E_{SFC}^K$ (Table 1), which can explain why $E_{SFC}^K$ is worst on the bias component. Finally, $E_{SFC}^P$ and $E_{SFC}^{KP}$ contain two SFCs for the timing of flows, while $E_{SFC}^K$ contains only one, which can explain why $E_{SFC}^K$ is not as good as the
two others for the correlation coefficient.

### 4.6.2   Individual streamflow characteristics

The normalised errors for the 18 SFCs that are contained in the three tailored objective functions (Figure 8) shows that, overall, all six objective functions tend to produce the largest errors on the same SFCs, for example on fh7 (frequency of large floods), fh6 (frequency of moderate floods), or tl1 (timing of annual minimum flow); and the smallest errors on the same SFCs, for
example dh13 (variability in annual minimum 30-day mean flow) or ra2 (variability in flow rise rate). The prediction of the SFCs considering the frequency of flow events (fh6, fh7, fh9, fl2) is the most difficult with all six objective functions, while their duration (dh4, dh13, dh16, dl9) are amongst the least difficult to predict. For the magnitude of flow events, low flow events seem to be relatively easy to predict while the magnitude of average and high flow events is more difficult.

However, $E_{SFC}^K$ and $E_{KG}^{Q^{-1}}$ tend to show more variability than the other four objective functions in the ranking of the errors
across the 18 SFCs. For example, $E_{KG}^{Q^{-1}}$ shows clearly larger errors on SFCs related to high flow conditions (mh10, fh6, fh7, fh9, dh4) which can be related to the focus on low flows of the objective function, but also on some average flow conditions (ma26, ma41, ra7), and even on low flow SFCs (ml20 – baseflow ratio). This is also the case for mostly the same SFCs with $E_{SFC}^K$. Again, these two objective functions place more emphasis on low flows, and this seems to make them less suitable objective functions across a range of flow conditions. On the other hand, the emphasis on high flows in $E_{SFC}^K$ seems less detrimental to
its performance on low flow conditions. It does perform the worst on ml17 (baseflow ratio) and q85 (flow exceeded 85 % of the time), but with relative errors below 10 %.

Moreover, unlike the overall performance results found in subsubsection 3.4.1, a tailored objective function does not necessarily perform the best on all of the SFCs it contains. For example, $E_{KG}^Q$ outperforms $E_{SFC}^P$ on ma41 (annual mean daily flow) which was already noticed with the bias. Also, $E_{KG}^{Q^{-1}}$ outperforms $E_{SFC}^P$ on q85, which can be explained by the strong emphasis
$E_{KG}^{Q^{-1}}$ on low flows. Interestingly, a tailored objective function can outperform another one on SFCs it does not contain. Indeed, $E_{SFC}^P$ outperforms $E_{SFC}^K$ on ra2, even though it is only contained in $E_{SFC}^K$.

### 4.7   Trends on a large range of flow regime characteristics

Extending the number of SFCs characteristics examined, shows that $E_{KG}^{Q^{0.5}}$, $E_{SFC}^K$, and $E_{SFC}^P$ perform very similarly across the 156 SFCs and the nine percentiles of the flow duration curve (Figure 9), and are somewhat similar with $E_{KG}^Q$, except for
the magnitude and duration of low flow events, where $E_{KG}^Q$ produces larger errors. This implies that $E_{KG}^{Q^{0.5}}$ is a strong option for model calibration when the purpose is to predict a wide range of streamflow characteristics, i.e. across various aspects (magnitude, frequency, timing, etc.) and across various flow conditions (high, moderate, and low flows), since it performs almost as well as the best tailored objective functions. In contrast, $E_{KG}^{Q^{-1}}$ produces noticeably larger errors than any other

objective function on the maximum daily flow in each month (i.e. mh1 to mh12), on the mean annual maximum of a moving mean of a 1-, 3-, 7-, 30- and 90-day window (dh1 to dh5), or on the frequency of flood events of various intensities (fh1, fh5, fh6, fh8). At the same time, this objective function produces noticeably smaller errors on the mean annual minimum of a moving mean of a 1-, 3-, 7-, 30- and 90-day window (dl1 to dl5). Overall, $E_{KG}^{Q^{-1}}$'s stronger weight on low flow conditions does improve the predictions of SFCs for low flow events, to the detriment of the prediction for high flow events. This is also noticeable in the percentiles of the flow duration curve, with a close to monotonic increase in the error amplitude from the 99[th] percentile to the 1[st] percentile. However, $E_{KG}^{Q^{-1}}$ is the worst objective function for predicting the minimum daily flow in each month for the period October-February (ml10, ml11, mh12, ml1, ml2). This is because the magnitude of low flows during this wet period are higher than during the dry period, so that errors for low flows for the dry period are given higher weight than the ones for the wet period in $E_{KG}^{Q^{-1}}$. Also, it has a larger error for predicting the frequency of low flow spells (fl1), that can be explained by the fact that the threshold to define low flow spells is set as the 25[th] percentile, which is not the magnitude of flows that is the most emphasised by $E_{KG}^{Q^{-1}}$ (i.e. not on the low tail of the flow distribution).

Amongst the tailored objective functions, $E_{SFC}^{K}$ performs differently across the 156 SFCs and the nine percentiles than its two counterparts, which perform very similarly across these SFCs. Indeed, $E_{SFC}^{K}$ shows absolute normalised errors somewhat half way between $E_{KG}^{Q^{-1}}$ and the two other tailored objective functions. $E_{SFC}^{K}$ tends to show larger errors on the characteristics where $E_{KG}^{Q^{-1}}$ is outperformed by the other traditional objective functions, typically on characteristics for low flow conditions. This pattern was already observed on the smaller set of SFCs in the subsubsection 4.6.2.

The relative agreement between the six objective functions in the largest and smallest SFC errors (apart from the patterns identified above), subsubsection 3.4.1 provides some insight on the most easy and most difficult SFCs to predict in the study catchments. It is clear that the average number of flow reversals from one day to the next (ra8) is the most difficult to predict correctly, and so are the average slope of the rising limbs and the recession limbs (ra1 and ra3) but to a lesser extent. Overall, high flow events are trickier to get right, whether it is their magnitude (mh1-mh12 – mean daily maximum for each month, mh19 – skewness in annual maximum daily flow, mh20 – mean annual maximum daily flow), their duration (dh1-dh10 – mean and variability in annual maximum of a moving mean of a 1-, 3-, 7-, 30- and 90-day window), their timing (th1 – timing of annual maximum flow), or their frequency, except for the variability in high flood events (fh2) and the average number of days exceeding seven times the median flow (fh4). On the other hand, some SFCs based on the magnitude of flows seem easier to predict: variability in the percentiles of the log-transformed discharge record (ma4), the skewness in daily flows (ma5), various ratios of flow percentiles (ma6-ma8) and various spreads between flow percentiles (m9-m11). The volume of floods exceeding the median, twice the median, and three times the median (mh21, mh22, and mh23) are also well predicted, alongside the 90[th] and 75[th] percentiles normalised by the median flow (mh16, mh17). Finally, the mean annual maximum of a moving mean of a 7-, and 30-day window normalised by the median flow (dh12, dh13) are the best predicted SFCs relating to the duration of flows. For the percentiles of the flow duration curve, it appears that all six objective functions are better suited to predicting its low tail, which is consistent with the lower relative errors for the SFCs on the magnitude of low flows compared with those of high flows.

## 5 Discussion

### 5.1 On the definition of SFC-based objective functions for ecologically-relevant streamflow predictions

The choice of the objective function for ecological applications is known to influence the predictive performance of the hydrological model for specific streamflow characteristics (Vis et al., 2015; Kiesel et al., 2017; Pool et al., 2017). In particular,
specially chosen composite objective functions containing the target SFCs have improved the prediction of these SFCs (e.g. Kiesel et al., 2017). This study confirmed these separate findings using the same set of SFCs in Irish study catchments. However, the consistency analysis revealed that the sample of parameter sets found suitable in calibration are less consistent across different split-sample tests with this type of objective function than with two of the traditional objective functions (i.e. $\mathrm{E_{KG}^{Q}}$ and $\mathrm{E_{KG}^{Q^{0.5}}}$).

As might be expected, because particular streamflow characteristics are selected for their ecological relevance does not imply that they are necessarily representative signatures of the overall hydrograph. Indeed, while some indicators originally used as ecologically-relevant SFCs (Olden and Poff, 2003) are also used as hydrological signatures (e.g. Yadav et al., 2007; Zhang et al., 2008), their selection as a relevant characteristic for the catchment of interest is driven by different needs in terms of the indicator skills. These indicators are selected as ecologically-relevant SFCs according to their influence on the stream ecology
(Poff and Zimmerman, 2010), while they are selected as hydrological signatures to represent the hydrological behaviour of catchments (McMillan et al., 2017), i.e. they are SFCs that can be used for catchment classification, or the regionalisation of hydrological information, for example. Hence, ecologically-relevant SFCs are not necessarily very informative when it comes to eliciting suitable parameter values in the calibration of hydrological models, because they may not be key descriptors of the emergent hydrological processes at the catchment scale: this may be symptomatic of the problem of getting the right answer
with a model for the wrong reasons (Kirchner, 2006). For example, Pool et al. (2017) defined a composite objective function made of the most informative SFCs at hand (i.e. the ones that, used alone, were the most useful to predict the other SFCs well too), and yet, they were unable to accurately predict SFCs not included in the objective function with their multi-SFC objective function. The use of a consistency analysis in this study confirms that the tailored objective functions tested are not skilled in selecting parameter values stable across split-sample tests. Nonetheless, some SFCs can be found useful in calibration. Yadav
et al. (2007) suggest that a carefully selected subset of SFCs has the potential to constrain well a model parameter space. Kiesel et al. (2017) even found that the use of single SFCs may be almost as powerful as their complete set of seven SFCs to predict all seven SFCs, suggesting that these individual ecologically-relevant SFCs also have potential to be indicative signatures of the hydrograph of their German catchment.

In this context, the definition of a good tailored objective function for ecologically-relevant streamflow predictions must be
based on SFCs that are key descriptors of the ecological response, while also key descriptors of the hydrological behaviour in the catchment. Otherwise, model consistency may be compromised, and the model predictions will not be as robust outside its calibration conditions. Moreover, the number of SFCs contained in the tailored objective function is another aspect that may need to be considered, given that the consistency seems to improve with the number of SFCs the objective function contains. However, only three set of SFCs were tested in this study, and more research would be required to confirm this hypothesis.

## 5.2 On the strengths of traditional objective functions

Composite traditional objective functions such as the Kling-Gupta efficiency remain strong contenders for the prediction of these SFCs. In particular, the use of the KGE on square-rooted flows (i.e. $E_{KG}^{Q^{0.5}}$) was as competitive as tailored objective functions consisting of specific SFCs to predict a set of 156 SFCs, while providing more robust predictions and a more consistent set of behavioural parameter sets than its tailored counterparts. On the other hand, $E_{KG}^{Q^{-1}}$'s stronger focus on low flow errors reduces its ability to predict SFCs for high flow events, which is a disadvantage, unless the ecological species of interest are only sensitive to low flow conditions. Even then, Garcia et al. (2017) found an arithmetic mean of $E_{KG}^{Q^{-1}}$ and $E_{KG}^{Q}$ better than $E_{KG}^{Q^{-1}}$ alone to predict low flow SFCs. Conversely, $E_{KG}^{Q}$'s heavier emphasis on high flow errors is not as detrimental for its predictive capabilities of low flow events, and is is only found marginally worse than $E_{KG}^{Q^{0.5}}$.

In future research on the skills of objective functions to predict SFCs, a recently formulated non-parametric version of the KGE criterion could prove useful to predict various SFCs at once, namely because it reduces the emphasis on high flow conditions and it provides a more balanced criterion across various flow conditions, while avoiding the original KGE's assumptions on the nature of the errors not necessarily justified for streamflow records (Pool et al., 2018). Alternatively, segments of the flow duration curve have been used to calibrate hydrological models, which also offers opportunities to balance low, average, and high flow conditions (e.g. Yilmaz et al., 2008; Pfannerstill et al., 2014). However, the flow duration curve does not contain information on the timing (or duration) of individual flow events, which is important for aquatic species (Arthington et al., 2006). A combination of different objective functions fitted to flows (Vis et al., 2015), or a combination of objective functions fitted to flows and objective functions fitted to SFCs (Pool et al., 2017) can also be competitive options. In particular, the latter has the potential to overcome the consistency issue found with tailored objective functions by including traditional objective functions.

## 5.3 Limitations of this study

The lack of long continuous time series of observed streamflow is known to be a limiting factor for ecohydrological studies, and, in this case study, the use of 14 years, i.e. 7 each for calibration and evaluation periods is a prime example of this issue. Previous research suggests that a five year period is enough to capture the temporal hydrological variability (Merz et al., 2009). However, Kennard et al. (2009) found that at least a 15-year period was required to estimate accurately a set of 120 SFCs, where the true SFC values were taken from their full record of 75 years. This suggests that the SFC values targeted in calibration in this study may not be fully representative of the long-term hydrological regime, and they are likely to be more variable than if 15-year (or more) periods were used in calibration and in evaluation, and hence more difficult to predict than more long-term values. Indeed, Vigiak et al. (2018) found that the uncertainty of the prediction of SFCs is sensitive to the length of the period considered, and the shorter the period is, the more uncertain the estimation is. Moreover, shorter time series reduce the likelihood of encompassing extreme flow events (droughts and floods).

Moreover, in order to overcome the lack of long time series of streamflow data, we included non-continuous (i.e. interrupted) data periods to increase the number of study catchments (see Figure A1). Given that missing discharge data tend to be more

frequent for high flows, because of flood events damaging the gauge, there is a risk that the natural flow variability is underestimated, and as a consequence the observed SFC values for extreme flow conditions may be less representative of the long-term flow regime than the SFC values for more moderate flow conditions. In our study, some hydrological years were discarded even if only one day of observation was missing. In future research, this requirement could be relaxed and imputation methods could be used on gaps of short length to infer the values for the missing days in the streamflow series (see, e.g. Gao et al., 2018, for a recent review of such imputation methods).

Given forcing and evaluation data uncertainty, and model structural uncertainties, the small differences in model performance calibrated with the different objective functions could be deemed insignificant. However, in order to reduce the influence of data uncertainty, this comparison of objective functions was carried out on a set of 33 study catchments and on 14 split-sample tests. Moreover, the use of the median performance across a set of behavioural parameter sets was chosen to reduce the influence of equifinality problems (Beven and Freer, 2001). Given that summary statistics across the split-sample tests and across the study catchments are used, this may explain why differences in performance are small. Regardless, the differences in terms of model robustness and consistency are more significant, and given the experimental set-up described above, this gives some confidence in the general applicability of these findings.

The findings in this study could also be somewhat model-specific and region-specific. However, Caldwell et al. (2015) found that the choice of the hydrological model to predict SFCs is not as important as the choices on the calibration strategy, and this study confirms the results of two other similar studies (Kiesel et al., 2017; Pool et al., 2017) that tailored objective functions perform better than traditional ones. In addition, the model suitability for the study catchments could be further explored following the covariance approach recently suggested by Visser-Quinn et al. (2019), and potentially improve on the model consistency.

Finally, the analysis of the consistency was based on the number of times the exact same parameter set was identified as behavioural across the 14 split-sample tests. However, it is possible that in some split-sample tests, a parameter set identified as behavioural is in the vicinity of another parameter set identified as behavioural in another test. This is one limitation of the consistency approach selected here, and it is suggested that future research efforts on the topic could use clustering analysis techniques in order to overcome this limitation by comparing the spread of the cluster(s) formed by the behavioural parameter sets instead.

## 5.4 Implications for the study of the impacts of climate change on the stream ecology

Hydrological models are usually preferred over statistical regression models when the impacts of a changing climate on the flow regime and the associated ecologically-relevant SFCs is of interest. Even though regression models may fit historical data better (Murphy et al., 2013), hydrological models have the potential to be run with alternative climate data in order to predict future changes in the catchment hydrograph. The identification of the most suitable objective function is therefore valuable for climate change scenario analysis. Here, we have already established the marginal superiority of tailored objective functions over a range of fourteen different split-sample tests in which the ranking between the objective functions is relatively stable. However, a limitation of the study is that the flow data period from 1986 to 2016 is relatively short in climatological terms

and does not contain a severe drought period, although some have been identified in long-term (250-year) precipitation records (Noone et al., 2017), but a corresponding flow record does not exist.

Assuming a suitable set of SFCs has been found, as described in subsection 5.1, the use of a composite definition for the objective function based on normalised absolute error between observed and simulated SFCs, such as in this study, or the recent studies by Kiesel et al. (2017), and by Pool et al. (2017) may not be realistic for practical applications. Indeed, while SFCs are often normalised to avoid artificially weighing them based on their amplitude, they are not weighed according to the impact a given percentage deviation has on the stream ecology. The use of an objective function whose components are weighted according to their significance to the target species may therefore prove useful to include such consideration in the calibration procedure. For example, Visser-Quinn et al. (2019) used variable limits of acceptability for the identification of the plausible model parameter sets based on a weighing scheme considering the importance of each of their SFCs on the ecological response, using macro-invertebrates as a surrogate (Visser et al., 2018).

## 5.5 Implications for ecologically-relevant streamflow predictions in ungauged basins

Understanding the ecological response to altered flow regimes is hindered by the lack of hydrological data where ecological data is available (Poff et al., 2010) because hydrometric gauges may not be in locations where ecological surveys have been carried out. As a result, the usual calibration of a hydrological model is not possible, and a direct method of predicting streamflow characteristics in ungauged locations is required.

One approach to regionalisation is the transfer of optimised parameter values from gauged to ungauged locations (Parajka et al., 2005). Given their higher consistency demonstrated in this study, the original KGE-based criteria appear better suited for regionalisation, rather than the tailored objective functions tested in this study. Indeed, the optimised parameter values need to be strongly related to catchment behaviour in order for hydrological knowledge to be related to physical features and thus transferred to ungauged locations. While consistency could be improved through the change in model structure (Euser et al., 2013), Caldwell et al. (2015) and Garcia et al. (2017) found the choice of the calibration procedure more decisive than the model used for the prediction of SFCs.

Alternatively, streamflow characteristics can be directly transferred from gauged to ungauged locations (e.g. Yadav et al., 2007; Westerberg et al., 2014) and used as calibration information in the ungauged catchment. However, these SFCs are used as hydrological signatures to constrain the model parameter space, and as a result, their potential was assessed in order to predict the hydrograph in ungauged catchments. It remains to be explored whether these regionalised ensemble predictions can prove useful in predicting other SFCs relevant for ecological communities in ungauged catchments.

## 6 Conclusions

Desirable qualities for a useful objective function are that it performs well in evaluation, i.e. outside calibration, that its performance is independent of the calibration period, and that it consistently identifies the same parameter sets regardless of the study period, i.e. that it describes a consistent catchment hydrological behaviour. This study explored all these aspects for

six different objective functions intended to predict three combinations of streamflow characteristics that are assumed to be relevant for stream ecology. The study showed that: (i) tailored objective functions perform marginally better than traditional objective functions to predict all three combinations of SFCs on average, while proving to be less robust outside calibration; (ii) traditional objectives functions based on flows and square-rooted flows select more consistently the same parameter sets as behavioural across the split-sample tests than the three tailored objective functions made of SFCs; and (iii) the ranking of the six objective functions is not altered when considering their performance on a very large and diverse set of SFCs.

This study unveils that a gain in fitting performance for the SFCs may hide a loss in consistency in the behavioural parameter sets across the split-sample tests. This highlights that fitting ecologically-relevant SFCs well is not necessarily a guarantee of representing all the key hydrological processes (i.e. informative signature) defining the catchment response. Unless streamflow characteristics are proven to be both ecologically-relevant and an informative signature at once, carefully selected traditional objective functions fitted to flows are likely to remain preferable to predict ecologically-relevant streamflow predictions to avoid consistency issues.

*Code and data availability.* The rainfall and potential evapotranspiration daily datasets are available online from Met Éireann (2019). The streamflow observations are available online from Ireland's Environmental Protection Agency (2019), and from the Office of Public Works (2019). The source code of the SMART model is open source and accessible online (Hallouin et al., 2019). The source code for the tools used to calculate the streamflow characteristics and the traditional objective functions are also open source and accessible online (Hallouin, 2019a, b).

*Author contributions.* This work is part of the PhD research of TH at the UCD Dooge Centre for Water Resources Research, under the supervision of MB and FOL. TH developed the idea. TH collected the data and performed the model simulations. TH wrote the original draft and the final version of this manuscript. MB and FOL reviewed and edited the different drafts of this manuscript.

*Competing interests.* The authors declare no conflict of interest.

*Acknowledgements.* The authors would like to thank Ireland's Environmental Protection Agency (EPA) for their financial support to the project ESManage (2014-W-LS-5).

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

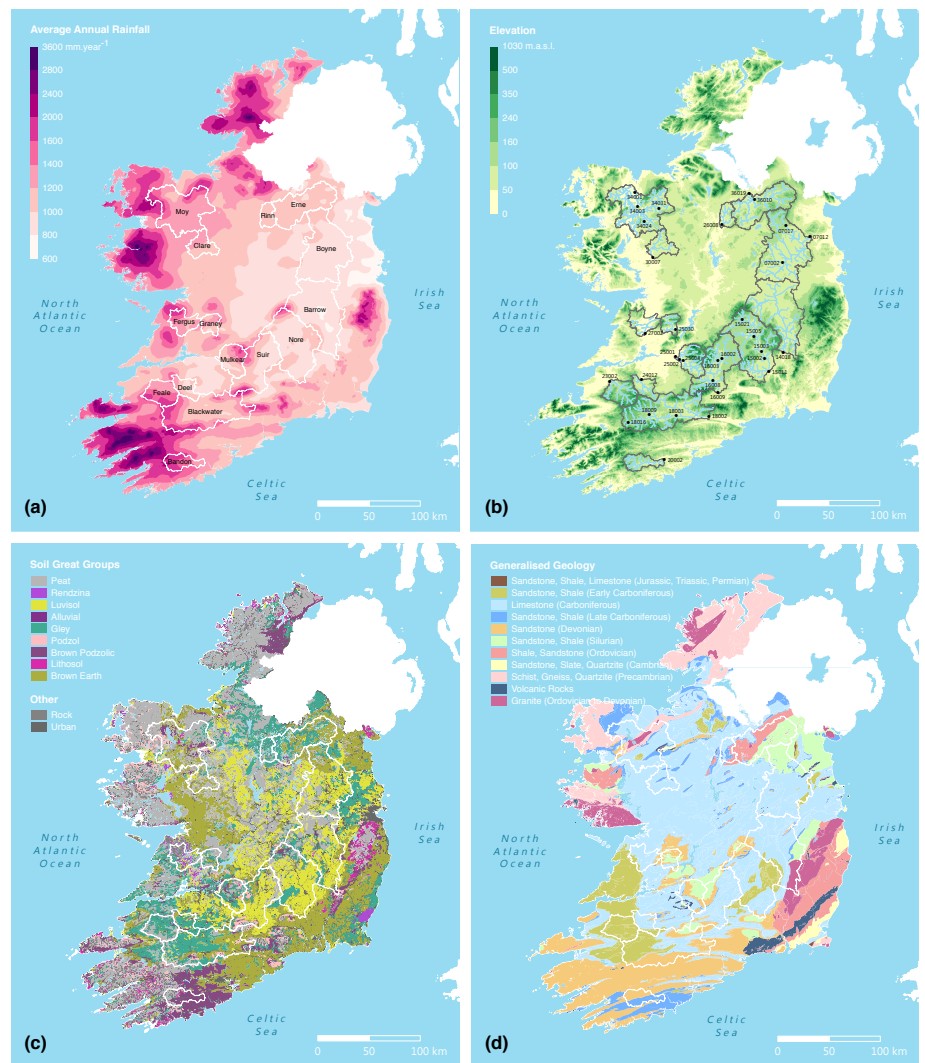

**Figure 1.** Spatial location and information on the study catchments: (a) map of the average annual rainfall for the Republic of Ireland for the period 1981-2010 (source: Met Éireann) overlaid with the 15 distinct river basins containing the 33 study catchments - each name corresponds to a river basin; (b) map of the topography for the Republic of Ireland (source: Ireland's EPA) overlaid with the location of the 33 hydrometric gauges forming the 33 study catchments - each number corresponds to the code of a hydrometric gauge; (c) map of the pedology for the Republic of Ireland (source: Teagasc) overlaid with the outlines of the 15 river basins; (d) map of the geology for the Republic of Ireland (source: Geological Survey Ireland) overlaid with the outlines of the 15 river basins.

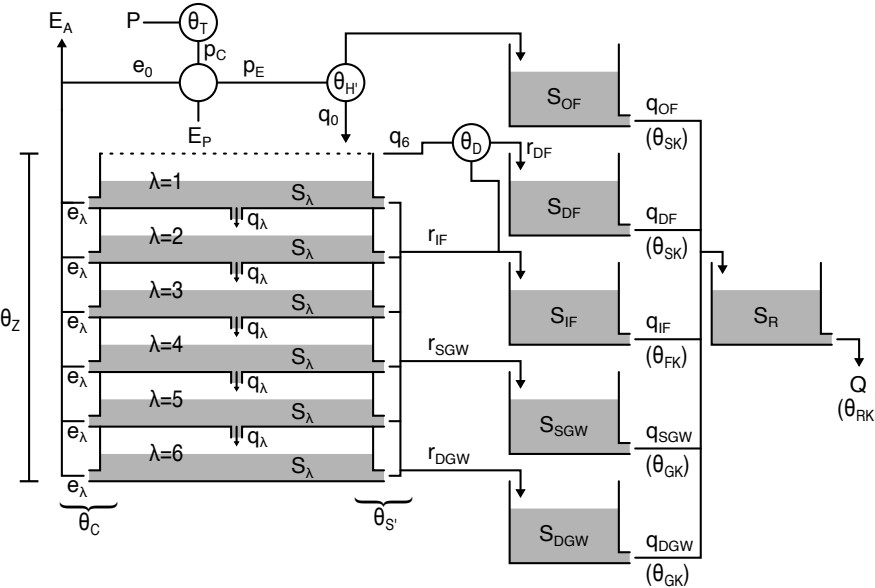

**Figure 2.** Conceptual representation of the SMART model structure. P and $E_P$, precipitation and potential evapotransporation, respectively, are the model inputs; Q and $E_A$, discharge and actual evapotranspiration, respectively, are the model outputs. For full description of the parameters, states, and fluxes presented on the figure, as well as the conceptual model equations, the reader is referred to the documentation provided in the Supplement.

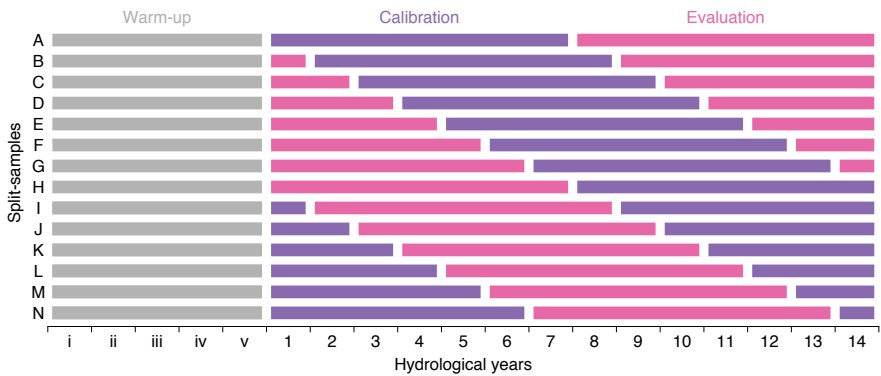

**Figure 3.** Split-sampling strategy using a seven-year rolling window, adapted from de Lavenne et al. (2016). Each period of 14 hydrological years enumerated as decimal numerals in Figure A1 are represented on the x-axis and split into two seven-year periods, one for model calibration (in purple), and one for model evaluation (in pink). Each period of 5 hydrological years identified as roman numerals on the x-axis corresponds to the 5-consecutive-year warm-up period immediately preceding the hydrological year number 1.

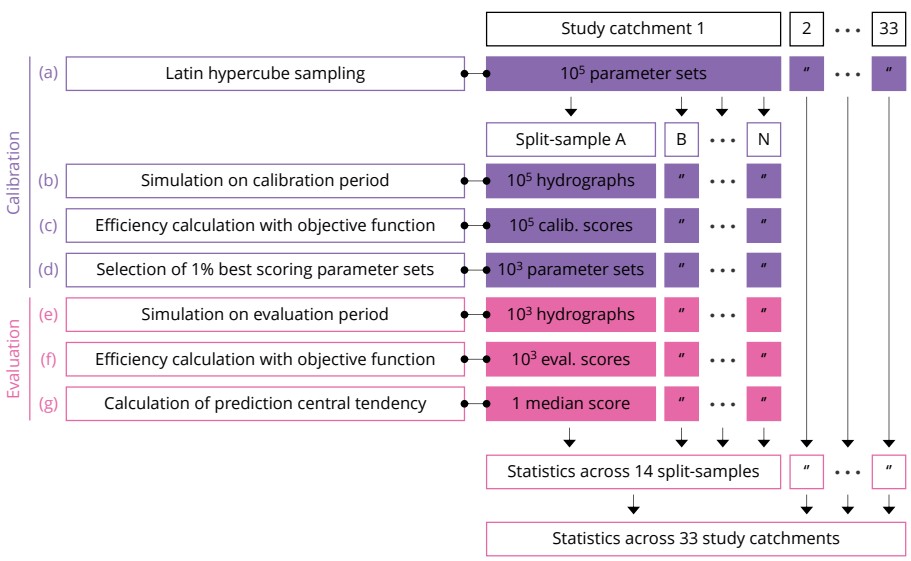

**Figure 4.** Model calibration and evaluation strategy for the prediction of SFCs with different objective functions. Steps (a) to (d) correspond to model calibration, while steps (e) to (g) correspond to model evaluation. These steps are replicated for each study catchment, and for each split-sample test.

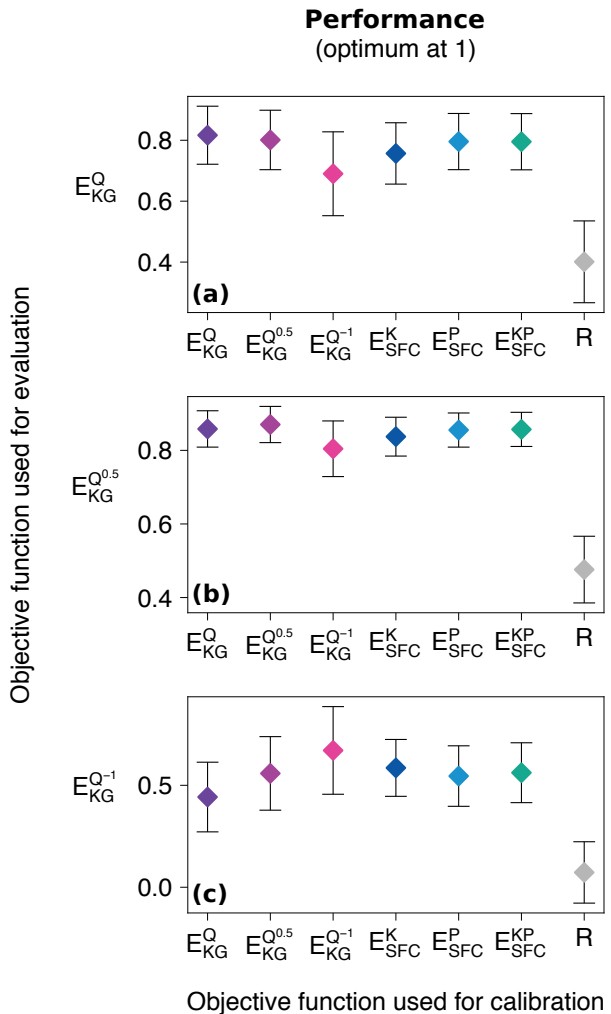

**Figure 5.** Comparison of the overall performance in evaluation of the model calibrated with the six objective functions. The three traditional objective functions used as evaluation efficiencies. $E_{KG}^{Q}$ corresponds to the original Kling-Gupta efficiency (Gupta et al., 2009).

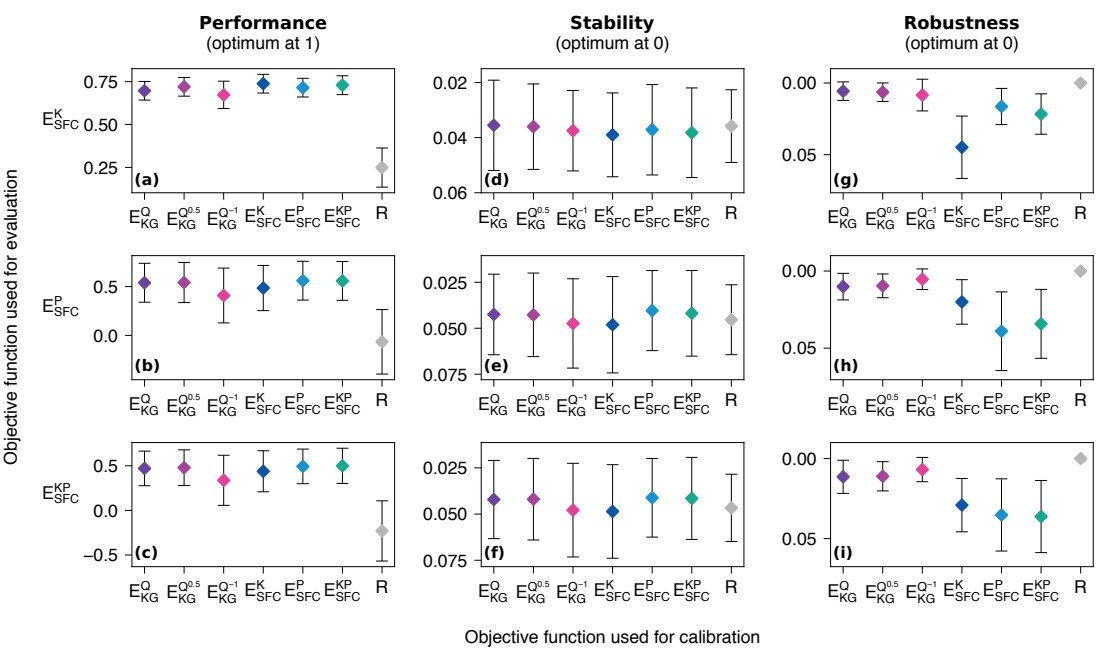

**Figure 6.** Comparison of the skills in evaluation of the model calibrated with the six objective functions. The first column of panels compares them on the overall performance on the three tailored objective functions used as evaluation efficiencies (described in subsubsection 3.4.1. The second column compares them on the stability of these efficiencies across the 14 split-sample tests (described in subsubsection 3.4.2. The third column compares them on the robustness of these efficiencies between calibration and evaluation periods (described in subsubsection 3.4.3).

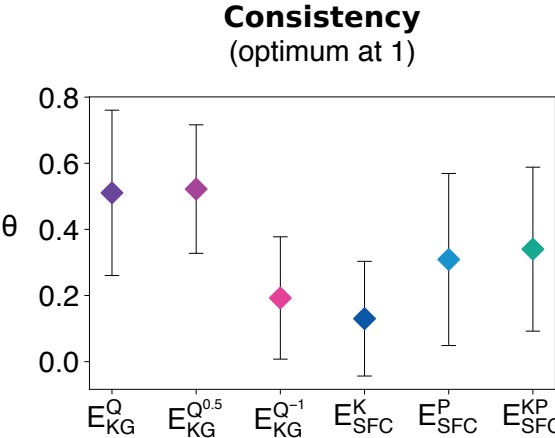

**Figure 7.** Comparison of the consistency of the set of behavioural parameter sets identified with the six objective functions across the 14 split-sample tests (described in subsubsection 3.4.4).

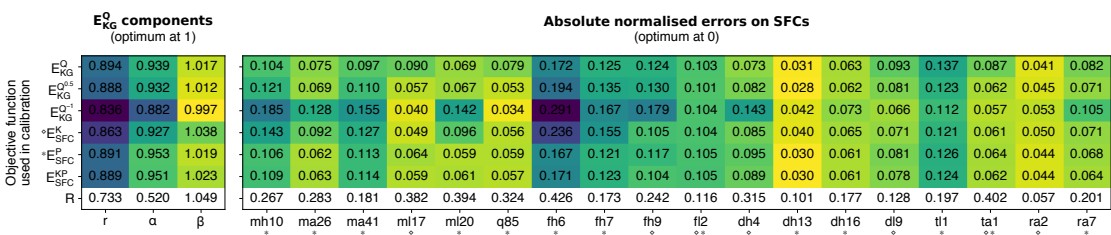

**Figure 8.** Comparison of performance in evaluation of the model calibrated with the six objective functions on individual components of the objective functions. The left panel compares them on the three component of the Kling-Gupta efficiency. The right panel compares them on the individual SFCs that are contained in the three tailored objective functions. The top panels correspond to performances on the calibration period, while the bottom panels correspond to the performances on the evaluation period. A diamond and an asterisk are used to display the SFCs belonging to $E_{SFC}^{K}$ and to $E_{SFC}^{P}$, respectively. Note, all SFCs belong to $E_{SFC}^{KP}$.

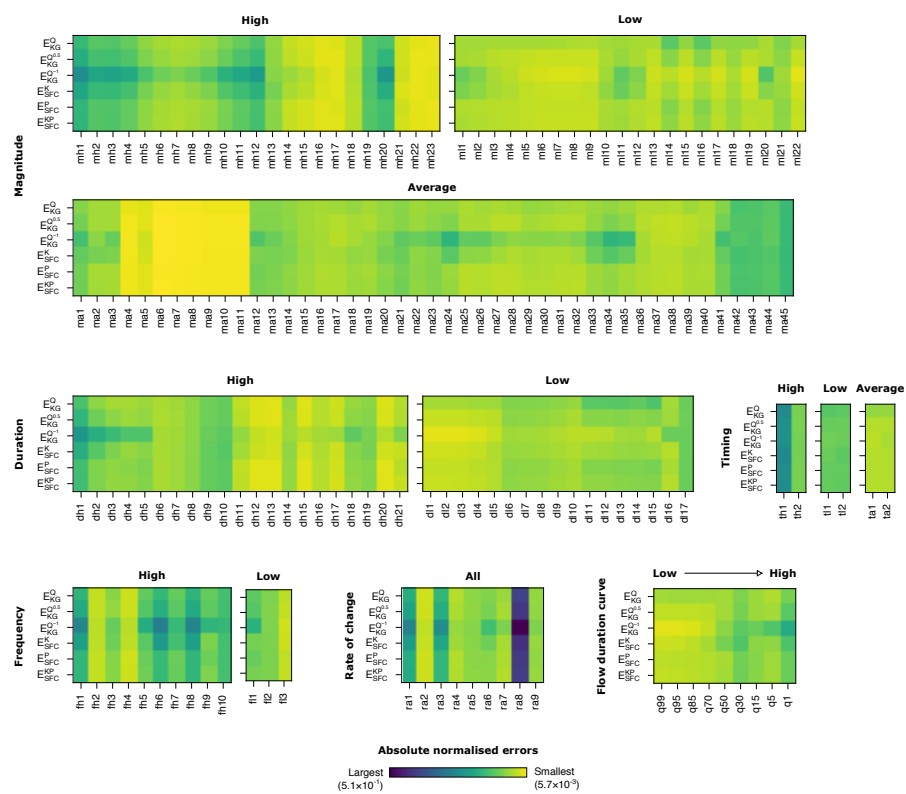

**Figure 9.** Comparison of performance in evaluation of the model calibrated with the six objective functions on 156 streamflow characteristics and 9 percentiles of the flow duration curve.

**Table 1.** List and description of the 18 selected streamflow characteristics. Detailed calculations for each SFC in available in Table A2. The three last columns indicate whether a given SFC is included ($\in$) or not included ($\notin$) in Equation 4 for the definition of each of the three tailored objective functions.

| Category | SFC | Description | Unit | $E^{K}_{SFC}$ | $E^{P}_{SFC}$ | $E^{KP}_{SFC}$ |
|---|---|---|---|---|---|---|
| **Magnitude** | | | | | | |
| Average flows | ma26 | Variability in March mean flow | % | $\notin$ | $\in$ | $\in$ |
| | ma41 | Annual mean daily flow | $m^3\,s^{-1}$ | $\notin$ | $\in$ | $\in$ |
| Low flows | ml17 | Base flow ratio 1 | − | $\in$ | $\notin$ | $\in$ |
| | ml20 | Base flow ratio 3 | − | $\notin$ | $\in$ | $\in$ |
| | q85 | Flow exceeded 85% of the time | $m^3\,s^{-1}$ | $\notin$ | $\in$ | $\in$ |
| High flows | mh10 | Mean October highest flood | $m^3\,s^{-1}$ | $\notin$ | $\in$ | $\in$ |
| **Frequency** | | | | | | |
| Low flows | fl2 | Variability in low flow pulse count | % | $\in$ | $\in$ | $\in$ |
| High flows | fh6 | Frequency of moderate floods | $yr^{-1}$ | $\notin$ | $\in$ | $\in$ |
| | fh7 | Frequency of large floods 1 | $yr^{-1}$ | $\notin$ | $\in$ | $\in$ |
| | fh9 | Frequency of large floods 2 | $yr^{-1}$ | $\in$ | $\notin$ | $\in$ |
| **Duration** | | | | | | |
| Low flows | dl9 | Variability in annual minimum 30-day mean flow | % | $\in$ | $\notin$ | $\in$ |
| High flows | dh4 | Annual maximum of 30-day moving mean flow | $m^3\,s^{-1}$ | $\in$ | $\notin$ | $\in$ |
| | dh13 | Variability in annual maximum 30-day mean flow | − | $\notin$ | $\in$ | $\in$ |
| | dh16 | Variability in high flow pulse duration | % | $\notin$ | $\in$ | $\in$ |
| **Timing** | | | | | | |
| Average flows | ta1 | Flow constancy | − | $\in$ | $\in$ | $\in$ |
| Low flows | tl1 | Timing of annual minimum flow | Julian day | $\notin$ | $\in$ | $\in$ |
| **Rate of change** | | | | | | |
| All flows | ra2 | Variability in flow rise rate | % | $\in$ | $\notin$ | $\in$ |
| | ra7 | Flow recession rate | $m^3\,s^{-1}$ | $\notin$ | $\in$ | $\in$ |

**Table 2.** List and Description of the ten parameters of the SMART model.

| Parameter | Description | Unit |
|---|---|---|
| T | Rainfall aerial correction factor | – |
| C | Evaporation decay coefficient | – |
| H | Quick runoff ratio | – |
| D | Drain flow ratio | – |
| S | Soil outflow coefficient | – |
| Z | Effective soil depth | mm |
| SK | Surface reservoir residence time | time step |
| FK | Interflow reservoir residence time | time step |
| GK | Groundwater reservoir residence time | time step |
| RK | Channel reservoir residence time | time step |

**Table A1.** List and main characteristics of the 33 study catchments.

| Hydrometric gauge | River basin | Drainage area[a] | Average rainfall[b] | Average potential evapotranspiration[b] | Baseflow index[c] | Mean elevation[d] | Mean slope[d] |
|---|---|---|---|---|---|---|---|
| – | – | $km^2$ | mm year$^{-1}$ | mm year$^{-1}$ | – | m | m km$^{-1}$ |
| 34031 | Moy | 25 | 1349 | 526 | 0.36 | 115 | 38.7 |
| 15021 | Nore | 70 | 1167 | 497 | 0.65 | 267 | 121.2 |
| 07017 | Boyne | 73 | 1016 | 501 | 0.55 | 147 | 68.9 |
| 18016 | Blackwater | 119 | 1660 | 526 | 0.35 | 211 | 54.2 |
| 34024 | Moy | 128 | 1217 | 526 | 0.52 | 82 | 34.8 |
| 25002 | Mulkear | 218 | 1342 | 572 | 0.54 | 192 | 97.5 |
| 16003 | Suir | 258 | 1485 | 568 | 0.57 | 154 | 65.9 |
| 25030 | Graney | 273 | 1301 | 570 | 0.55 | 135 | 74.7 |
| 07002 | Boyne | 286 | 981 | 503 | 0.78 | 96 | 23.0 |
| 26008 | Rinn | 297 | 1182 | 498 | 0.61 | 75 | 46.6 |
| 15003 | Nore | 299 | 1029 | 537 | 0.55 | 208 | 56.4 |
| 18009 | Blackwater | 311 | 1286 | 574 | 0.42 | 199 | 66.6 |
| 24012 | Deel | 366 | 1109 | 569 | 0.43 | 116 | 41.4 |
| 15005 | Nore | 380 | 916 | 499 | 0.71 | 127 | 28.6 |
| 25003 | Mulkear | 399 | 1183 | 568 | 0.50 | 140 | 64.9 |
| 20002 | Bandon | 422 | 1654 | 528 | 0.53 | 124 | 89.4 |
| 30007 | Clare | 476 | 1121 | 504 | 0.65 | 75 | 23.8 |
| 27002 | Fergus | 485 | 1497 | 574 | 0.67 | 74 | 53.3 |
| 16002 | Suir | 492 | 972 | 568 | 0.63 | 128 | 19.3 |
| 23002 | Feale | 647 | 1409 | 567 | 0.31 | 196 | 76.2 |
| 25001 | Mulkear | 648 | 1235 | 578 | 0.52 | 153 | 73.7 |
| 36010 | Erne | 762 | 1041 | 498 | 0.63 | 124 | 82.6 |
| 16008 | Suir | 1090 | 1145 | 572 | 0.64 | 138 | 41.6 |
| 18003 | Blackwater | 1255 | 1389 | 524 | 0.46 | 181 | 68.2 |
| 36019 | Erne | 1491 | 1048 | 498 | 0.79 | 107 | 73.4 |
| 16009 | Suir | 1586 | 1213 | 575 | 0.63 | 139 | 51.4 |
| 15002 | Nore | 1647 | 980 | 502 | 0.63 | 149 | 43.9 |
| 34003 | Moy | 1782 | 1406 | 527 | 0.79 | 82 | 48.4 |
| 34001 | Moy | 1961 | 1396 | 520 | 0.78 | 81 | 49.7 |
| 15011 | Nore | 2222 | 973 | 501 | 0.62 | 139 | 42.7 |
| 18002 | Blackwater | 2331 | 1308 | 526 | 0.62 | 166 | 70.3 |
| 14018 | Barrow | 2438 | 919 | 536 | 0.67 | 99 | 27.0 |
| 07012 | Boyne | 2462 | 930 | 502 | 0.68 | 91 | 26.5 |

Data sources: [a]EPA river sub-basins map, [b]Met Éireann weather stations, [c]OPW Flood Studies Update, [d]EPA digital terrain model

**Table A2.** Detailed computations for the 18 selected streamflow characteristics.

| SFC | Description |
|---|---|
| | Detailed calculations |

**ma26**    **Variability in March mean flow**

Compute the mean and standard deviation in daily flows in March for each hydrological year. Divide the standard deviations by the means. Calculate the mean of these ratios to get ma26.

**ma41**    **Annual mean daily flow**

Compute the mean daily flow for each hydrological year. Divide the means by the drainage area in square kilometers. Calculate the mean of these ratios to get ma41.

**ml17**    **Base flow ratio 1**

Compute the 7-day rolling mean for each hydrological year. Calculate the minimum rolling mean and divide by the mean daily flow for each hydrological year. Calculate the mean of these ratios to get ml17.

**ml20**    **Base flow ratio 3**

Break down the entire record of daily flows into 5-day blocks. Calculate the minimum flow in each block. This minimum is set as the baseflow for the block if 90% of its value is less than the minimum flow of its preceding and following blocks. Otherwise baseflow for this block is unassigned. Replace all unassigned baseflow values using linear interpolation on the already assigned baseflow values. Calculate the total baseflow by summing up the baseflow values in each 5-day block, and the total flow for the entire record. Calculate the ratio of these two totals to get ml20.

**q85**    **Flow exceeded 85% of the time**

Calculate the 15$^{th}$ percentile for the entire record to get q85.

**mh10**    **Mean October highest flood**

Compute the maximum daily flow In October for each hydrological year. Calculate the mean of these values to get mh10.

**fl2**    **Variability in low flow pulse count**

Calculate the 25$^{th}$ percentile for the entire record. Calculate the number of flow events that are below this percentile for each hydrological year. Calculate the coefficient of variation (i.e. standard deviation divided by mean) of these values and multiply by 100 to get fl2.

**fh6**    **Frequency of moderate floods**

Calculate the median for the entire record. Calculate the number of flow events that are above 3 times this median for each hydrological year. Calculate the mean of these values to get fh6.

**fh7**    **Frequency of large floods 1**

Calculate the median for the entire record. Calculate the number of flow events that are above 7 times this median for each hydrological year. Calculate the mean of these values to get fh7.

**fh9**    **Frequency of large floods 2**

Calculate the 25$^{th}$ percentile for the entire record. Calculate the number of flow events that are above this percentile for each hydrological year. Calculate the mean of these values to get fh9.

| SFC | Description |
|-----|-------------|
| | Detailed calculations |

**dl9**     **Variability in annual minimum 30-day mean flow**

Compute the 30-day rolling mean for the entire record. Calculate the minimum of this rolling mean for each hydrological year. Calculate the coefficient of variation (i.e. standard deviation divided by mean) of these values and multiply by 100 to get dl9.

**dh4**     **Annual maximum of 30-day moving mean flow**

Compute the 30-day rolling mean for the entire record. Calculate the maximum of this rolling mean for each hydrological year. Calculate the mean of these values to get dh4.

**dh13**     **Variability in annual maximum 30-day mean flow**

Compute the 30-day rolling mean for the entire record. Calculate the maximum of this rolling mean for each hydrological year. Calculate the mean of these values and divide by the median daily flow for the entire record to get dh13.

**dh16**     **Variability in high flow pulse duration**

Calculate the 75$^{th}$ percentile for the entire record. Calculate the average duration of flow events above this percentile for each hydrological year. Calculate the coefficient of variation of these values and multiply by 100 to get dh16.

**ta1**     **Flow constancy**

Decimal log-transform the entire record of daily flows. Calculate the decimal log of the mean daily flow for the entire record. Compute the Colwell (1974) matrix featuring 365 rows for 365 days in a year (ignoring last day of February for leap years) and 11 columns for 11 flow states (break points are 0.10, 0.25, 0.50, 0.75, 1.00, 1.25, 1.50, 1.75, 2.00, and 2.25 times the log mean daily flow calculated previously) for each hydrological year, incrementally adding to the tally in each cell from year to year. Calculate the sum of each column Y (vector), and the sum of the whole matrix Z (scalar). Divide the elements of vector Y by scalar Z. Multiply the elements of the new vector by their respective decimal log-transformed value, sum the elements of the vectors to obtain a scalar and multiply by minus one to obtain the uncertainty with respect to the states H(Y). Divide H(Y) by the decimal log of the number of states (11), and subtract this ratio from one to get ta1.

**tl1**     **Timing of annual minimum flow Julian day**

Determine the date of the annual minimum daily flow in the Julian calendar for each hydrological year. Convert these values into an angle in the unit circle. Compute their coordinates (i.e. cosine and sine). Calculate the mean of these two values separately. Calculate the ratio of this mean sine divided by this mean cosine. Calculate the arc tangent of this ratio to get the angle corresponding to these mean coordinates. Convert this angle back to a Julian date to get tl1.

**ra2**     **Variability in flow rise rate**

Compute the difference in daily flows between each consecutive days for the entire record. Calculate the coefficient of variation (i.e. standard deviation divided by mean) for the positive differences (i.e. rising limbs) and multiply by 100 to get ra2.

**ra7**     **Flow recession rate**

Natural log-transform the entire record of daily flows. Compute the difference in this log-transformed daily flows between each consecutive days for the entire record. Calculate the median of the negative differences (i.e. recession limbs) to get ra7.

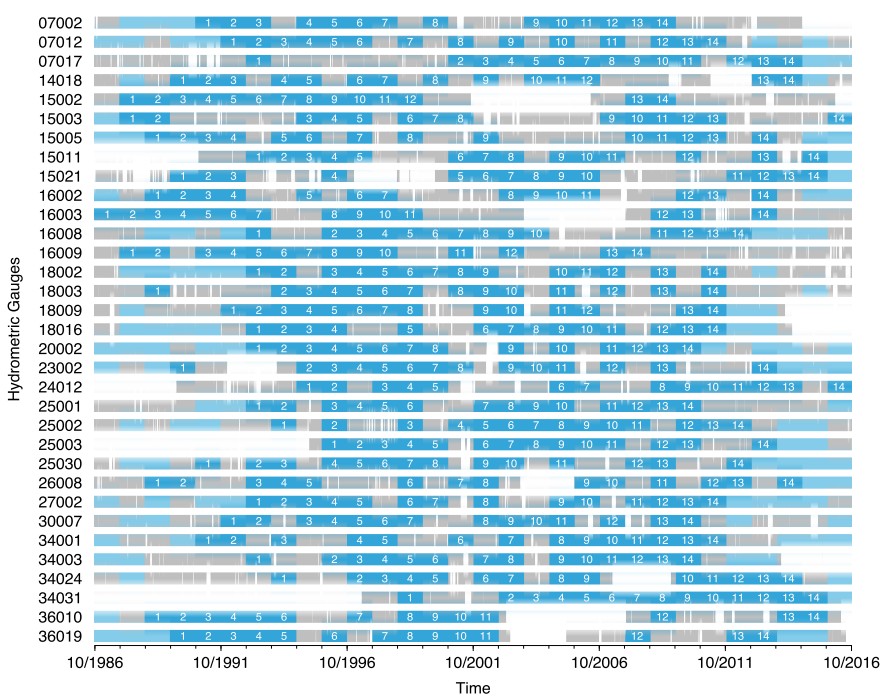

**Figure A1.** Discharge data availability for the 33 study catchments. The 14 complete hydrological years selected are represented in dark blue and annotated from 1 to 14. Years in light blue are other complete hydrological years not retained. Discontinuous grey years contain missing data represented as a discontinuity in the bar.