# Peer review of "Calibration of hydrological models for ecologically-relevant streamflow predictions: a trade-off between fitting well to data and estimating consistent parameter sets?"

_Hydrology and Earth System Sciences, 2019_

## Referee Comment (RC1) · Anonymous Referee #1 · 3 Jun 2019

**Review of the manuscript "Calibration of hydrological models for ecologically-relevant streamflow predictions: a trade-off between performance and consistency" by Hallouin et al.**

Hydrological models are an important tool for evaluating the effect of altered runoff regimes on the ecological integrity of a freshwater system. However, the accurate prediction of multiple specific hydrograph aspects (SFCs) from a single streamflow simulation is challenging. Hallouin et al. address this challenge by evaluating model simulations from calibrations with different objective functions. The selected objective

functions are the Kling-Gupta efficiency (and variants) and some objective functions consisting of ecologically relevant SFCs. Using 14 split-sample tests with a moving window, model performance, performance stability, and consistency in the selection of parameter sets are evaluated. Results from simulations in 33 Irish catchments indicate that model performance for SFCs tends to be higher when parameter sets are selected using SFC-based objective functions. However, SFCs based objective functions tend to have a lower consistency than traditional objective functions such as KGE.

This study addresses the current challenge of calibrating hydrological models to make accurate and robust predictions of multiple SFCs. The combination of performance, stability and consistency makes the results of this study especially valuable and highly relevant for the prediction of SFCs in places or times without data. I really like the concept of the three-fold model evaluation and I think it is presented in a clear well structured way. Generally, I think it would be important to do a more in-depth analysis of the results to improve the understanding of why there are differences in performance, stability and consistency. Also, I think that more references to related work would improve the scientific background of the research questions and could enhance the quality of the discussion.

I hope that the comments below will be helpful for the authors to improve their manuscript.

**General comments**

The differences in performance, stability, and consistency are evaluated in terms of KGEhi, KGEav, KGElo, Dinv, Dfsh, and Dall. These are all multi-objective criteria and their final value is an aggregated metric over multiple aspects of the hydrograph performance. Looking at the single components of KGE and also at single SFCs could potentially allow us to gain some insight into why differences in performance, stability,

and consistency can be observed (e.g. is the performance of a particular bespoke objective function lower than the one of KGEhi because of a few SFCs that are very poorly predicted?). I think it is especially the "why" that helps us to learn more about the effect of objective functions on simulations and that ultimately helps us to improve predictions of various hydrograph aspects.

Using the concept of consistency, this study looks at the ability of objective functions to consistently select the same set of behavioural parameters across multiple calibration time windows. Consistency is evaluated and compared across objective functions using a fraction-based score (i.e. fraction of parameter sets that is constantly behavioural). I think it could be interesting to also look at the parameter values themselves by e.g. plotting the values of the most sensitive parameters against model efficiency. Eventual patterns in parameter values might then be helpful to understand differences in model performance from different objective functions.

Sometimes differences in model performance between objective functions are rather small (as mentioned by the authors in section 4.2). Is it possible that these differences only seem to be small because model performance cannot get too bad in the study catchments? It might be worth to compare results against a benchmark, such as a random selection of 1

The study is based on three hypotheses addressing model performance, stability, and consistency. The introduction provides the reader mostly the background and motivation for hypothesis H1 (model performance). However, there is not much information about the current knowledge and experience on the stability of model performance and the consistency in parameter selection. I recommend to extend the introduction so that it covers the whole range of questions addressed in the study.

**Specific comments**

Abstract: More than half of the text in the abstract is introduction. I suggest to shorten

that part and provide more information on the methods and results of the three hypotheses of the study (performance, stability, and consistency).

Introduction: The studies of Kiesel et al. (2017) and Pool et al. (2017) are often cited as examples for using streamflow characteristics in model calibration. However, there are many more studies using hydrological signatures for model calibration. Although these studies don't aim at simulating ecological flow indices it might be worth to cite some of them.

P4 L 4-14: It makes clearly sense to me that you select the same SFCs as have been used in previous studies. However, I am not sure about the use of the term "ecologically relevant" in e.g. your title. To my knowledge, the relevance of SFCs depends on the species of interest. I wonder if the same species are important (or exist) in Ireland as in Germany and the Southeastern US? If not, can the selected SFCs still be considered as ecologically relevant?

P4 L20: Seven years are selected for calibration. Are seven years enough to get a robust estimate of the SFCs used in this study? I think it would be worth to discuss that and think about the consequences for the results on the stability of performance.

P4 L24-30: Can you say how many catchments are nested? How does that affect the generalization of the results? You mention that the catchments represent the diversity in soil types and geology - therefore, I would provide more explicit information about soil type and geology. Is there any snowfall in the study catchments?

P5 L26: Model set-up: How was potential evapotranspiration calculated? How was the warming-up period selected in case of the moving split-sample tests?

P6 L20: It is a common practice to not only transform flows to put more emphasize on low flows but also put more emphasize on mean flows (by e.g. calculating the sqrt of flows). What was the reason for calculating the mean of KGEhigh and KGElow instead of making a transformation?

P7 L12: Given that the focus of the paper is on the comparison of traditional and bespoke objective functions, I recommend to explicitly write down the equation for each of the three bespoke objective functions so that the reader doesn't have to check Table 2 to know which SFCs and how many are in each bespoke objective function.

P7 L12: The traditional objective functions (KGE variants) have their optimal value at 1 whereas the bespoke objective functions have an optimal value of 0. This can be very confusing when interpreting the results in Figs. 6, 7 and 8. I strongly recommend to define the bespoke objective functions in the same way as the KGE, i.e. 1 - Euclidean distance.

P7 L16: Model evaluation: A description of the concepts of stability and consistency is missing in the section on model evaluation. On the other hand, many sections of the results (P8 L 20-25; P8 L27-P9 L2; P9 L 25-L31; P10 L9-15) start with an extended paragraph on methods. I would move these paragraphs to the section on model evaluation to make a clear separation between methods and results.

P8 L 10-14. How were the 14 combinations of the 7-year periods chosen?

P11 L 4: There are more studies looking at the effect of objective functions on the prediction of streamflow characteristics than those of Vis et al. (2015), Kiesel et al. (2017), and Pool et al. (2017) (two examples are Hernandez-Suarez et al., 2018; Zhang et al., 2016)

Discussion: About half of the discussion is about the implication of the results for climate change studies and studies on the prediction in ungauged basins. However, climate change and prediction in ungauged basins are not directly addressed in this study. I agree that the results are interesting and relevant for these two topics and it is important that you do discuss the implication of your results for climate change and prediction in ungauged basins. But generally, I recommend to rearrange the discussion to focus much more on the findings and limitations of this study: which hydrological processes are represented by different objective functions? What causes

differences in performance and consistency? Why is there not much difference in the stability of model performance?

**Detailed comments**

Title: Given that the results don't show very strong differences between performance and consistency between traditional and bespoke objective functions I would think about using a question mark at the end of the title.

P3 L29-32: H1: It is hypothesized that bespoke objective functions lead to a better model performance than traditional objective functions. This is a rather vague formulation and I would explicitly state that it is about a better model performance. H2: What is a "small" number of SFCs? Again, I would be more explicit in the formulation of the hypothesis.

Fig. 2: I agree that the information in Fig. 2 is interesting for someone working with the same data set. But its information is maybe not so relevant for the general reader. You could think about placing the figure in the supplemental material.

Fig. 4: i) For me step f and g go hand in hand and I would merge them as you do with the calculation of the objective function in step c. Is there a reason why you use once the term "mean" and once the term "average"?

P6 L30: I would suggest to shortly explain how exactly flows were normalized.

Fig. 5: I am not sure if this figure is needed. The concept of the split sample test with a moving window is already well explained in the text.

Fig. 7: Are the axis labels in 7a and 7b switched, i.e. should 7a be Dinv and 7b be Dfsh?

Fig. 7: You often use the term "Euclidean distance" when talking about the SFCs based objective functions. These objective functions are defined in section 3.2 and I don't think it is necessary to repeat that they are based on the Euclidean distance.

P9 L29: "... more holistic definition that traditional objective functions represent...". I think this is a delicate statement and needs some explanation, especially since KGE consists of three components and the bespoke objective functions consist of many more components.

P10 L27: I think it is rather the model performance that is consistent than the catchments themselves.

**References**

Hernandez-Suarez, J. S., Nejadhashemi, A. P., Kropp, I. M., Abouali, M., Zhang, Z., Deb, K. (2018). Evaluation of the impacts of hydrologic model calibration methods on predictability of ecologically-relevant hydrologic indices. Journal of hydrology, 564, 758-772.

Zhang, Y., Shao, Q., Zhang, S., Zhai, X., She, D. (2016). Multi-metric calibration of hydrological model to capture overall flow regimes. Journal of Hydrology, 539, 525-538.

---

## Referee Comment (RC2) · Anonymous Referee #2 · 11 Jun 2019

Review of the manuscript "Calibration of hydrological models for ecologicallyrelevant streamflow predictions: a trade-off between performance and consistency" by Hallouin et al.

In this paper, Hallouin et al. describe trade-offs in the choice and formulation of objective functions used to predict streamflow characteristics (SFCs) from rainfall-runoff models. Using simulations from 33 catchments in Ireland, the authors evaluated the overall performance, stability, and consistency of 6 objective functions, consisting of 3 "traditional" functions (variants of Kling-Gupta efficiency) and 3 "bespoke" functions

(based on SFCs identified in two sets of studies from the southeastern U.S. and Germany).

Overall, the paper is well-written, structured in a logical format, and addresses relevant questions for the task of calibrating process-based (e.g. rainfall-runoff) models to predict suites of multiple SFCs. However, I agree with reviewer #1 that a more in-depth assessment of the reasons for variability in objective function performance, stability, and consistency would improve the paper. Currently, SFCs are organized into "fish" and "invertebrate" categories, which I feel is potentially misleading (as described below). It could be more instructive, perhaps in future work, to organize SFCs instead based on the types of information they represent (e.g. timing and duration SFCs versus magnitude SFCs, or SFCs that describe extremes such as high-flows and low-flows versus those that describe central tendency). Such an organization could provide more useful information as to the processes by which certain groups of SFCs are better or more poorly predicted by different objective functions. Absent such a reorganization in this paper, the authors could still probe a little more deeply into the types of SFCs in each grouping to explore how/why the reported patterns arise.

General comments:

The second paragraph of the introduction is a bit generic in referring to "a range of streamflow characteristics". Some examples could help: low-flow versus high flow periods and their associated SFCs? Timing and duration statistics as opposed to magnitude statistics? It would help at this point in the introduction to give the reader a better sense of the types and categories of SFCs that are commonly used, and how and why different types of SFCs might be optimized using different functions. Different aspects of the flow regime are specifically named in methods (P4 L9-10), but would be better introduced and explained in the introduction.

The adjective "bespoke" may be unfamiliar to some audiences, e.g. in the USA. Is there another term that could be more globally well understood? Perhaps "customized"?

Also, can you provide a succinct definition of "behavioural" parameter and how parameters are identified as "behavioural"?

Fig. 2 caption indicates a substantial number of complete hydrological years were discarded from analysis (light blue) but this is not explained in methods. Why was this done? Was it so as to have an equal number of years of available data across catchments, and the minimum available was 14 years? Or was this to facilitate the split-sample tests? Were there trade-offs in this decision, i.e. loss of precision due to using less calibration or validation data than was actually available?

Throughout the text (e.g. P4 L8 and in Table 2), sets of SFCs are presented as belonging to "fish" or "invertebrates" when really they represent SFCs drawn from two sets of studies—including Knight et al (2014) (fish) and Kakouei et al (2017) (invertebrates)—that happened to use these taxonomic groups. As currently presented, this gives the impression that these sets of SFCs have empirically been demonstrated to be more important to fish than to inverts, or vice versa. Similarly, labeling the vectors and corresponding Euclidean distances as "Dfsh" and "Dinv" gives this impression. The cited studies from which these SFCs were derived were not focused on the relative importance of SFCs to one taxonomic group over another. To truly represent such broad taxonomic groups, studies across a wider set of geographic areas (not just the SE United States and Germany) and more diverse sets of species would be required. Regardless, this is not a goal of this paper. This could be presented differently, with the same SFCs and same analysis, in a way that would not give false impressions about overall taxonomic relevance for particular groups of SFCs.

There are some problems of disagreement between results text and results figures. For example, the results text in section 4.2 does not match the results shown in Fig. 7 (see specific comments below). Please ensure correct labeling of figure panels and revise text or figure as needed to ensure agreement. There is also disagreement between results text in section 4.4. and results shown in Fig. 9 (see specific comments below).

[Figure]

In the results as presented in section 4.3 and Fig. 8, stability of performance across the split-sample tests is apparently assessed only visually. Even for this study with a relatively small number of catchments (33) this becomes a bit unwieldy (and the reader is forced to search through 33 separate plots in each of 3 figure panels to find the examples mentioned in the text). If these methods were to be reproduced or adapted for another study area with a larger number of catchments (in the hundreds, as is common), then a figure such as Fig. 8—upon which the conclusions about stability rely—would become untenable. This paper would benefit from a more quantitative assessment of stability, presented perhaps as means and standard deviations across catchments instead of Fig. 8. Such a quantitative stability metric would (a) be much easier to understand and interpret, while presumably yielding the same results, namely that performance stability was not markedly different between traditional and bespoke objective functions, and (b) would be more repeatable and transferable to other studies in other regions, especially using large numbers of catchments.

In conclusions, I'm not sure that hypothesis 3 is strongly supported by results in Fig 9 (see specific comments below).

Table 2 is cited before Table 1 (section 2.1 and 2.3, respectively). Likewise, Fig. 1 is cited after Figs 2 and 3. Please ensure all tables and figures are cited in proper order.

Specific comments:

P2 L5: "The prediction of streamflow conditions" [need to add or specify "at ungauged locations"]

P2 L7 "hydrological models that produce streamflow hydrographs" [may want to specify these are simulated hydrographs as opposed to observed hydrographs]

P2 L15 "across a range of streamflow characteristics"... does this particularly involve high-flow vs low-flow periods?

P2 L16-17: sentence fragment. Delete "although"?
P3 L9: is there a missing comma? "captured in the effective parameter values of the model, could be compromised..."

P3 L12-12: "...relating to its own preferences for living conditions" this was also demonstrated by Knight et al (2014).

P3 L13-14: "when several species are considered simultaneously, the number of SFCs to predict will increase accordingly"... yes, in general, but that depends on the habitat requirements of the species in question. Species with similar behavior (eg foraging strategies), reproductive timing, and physical (eg thermal) niches may actually share a common set of SFCs as being most relevant to them.

P4 L3-7: Any evidence that these SFCs are ecologically relevant in the Irish catchments?

P4 L8: "Only two hydrological indices are common between the two communities' respective streamflow preferences". I don't think—based solely on the cited studies in the Southeastern US and Germany—that such sweeping statements can be made about streamflow preferences across such broad taxonomic groups (fish and invertebrates). More accurately, this is a comparison between two sets of studies on two different continents that happened to use different taxonomic groups in their analysis. It does not support conclusions about which SFCs matter most to which taxonomic groups outside the respective study areas of the cited studies. I am familiar with the research in the Southeastern US (Knight et al 2014) which did not use any invertebrate data at all, and so should not be used to suggest that certain SFCs are more (or less) important to fish relative to inverts in that study region.

P4 L13: "calculations were vectorised"... can you explain a bit more what this means? I'm familiar with EflowStats but not Python. Perhaps Python users know exactly what you mean, but a brief explanation could help readers with less familiarity. In Table 2, the column header "target species" (fish or invertebrates) might give a casual reader the impression that these SFCs were empirically deemed important in the study-area

catchments. Rather, they were gleaned from Knight et al (2014) (fish) or from Kakouei et al (2017) (invertebrates). More importantly, it could give a false impression that these SFCs are generally more important to the taxonomic group listed which is not necessarily the case based on the previous studies cited. This could be clarified in the table caption or the accompanying text, or by changing "target species" to "citation" and listing the corresponding paper from which the SFC was obtained. Also in Table 2, table caption says "SFC" but column header says "indicator". Would be better to use one term consistently for clarity. This goes for the main text as well, where "SFCs" and "hydrologic indices" are both used.

P4 L20: "in order to have at least seven years for calibration and seven years for evaluation. . ." Can you justify (perhaps with citation) why seven years is an acceptable POR for calibration/validation?

P4 L27-28: "hence representing a good sample of the diversity of Irish soils and geology" Except that from Fig 3 it appears that higher-elevation or mountainous catchments were not well represented. . .? Fig 1 and Fig S1 are essentially redundant to each other, and Table 1 and Table S3 are redundant, except for slight differences in parameter presentation. Parameter representations in Table S3 appear to match Fig S1 but not Fig 1. Suggest retaining only one version of this figure/table combination, or if repeating in the supplement then ensure consistent parameter representations throughout.

P5 L14 and L19: How are energy-limited and water-limited periods defined?

P5 L27: What formulation of ET is this? E.g. Penman-Monteith?

P6 L23: do these KGE variants really encompass "the entirety of the observed and simulated hydrographs"? It seems the first KGE emphasizes high flows, the second (inverted) emphasizes low flows, and the third "equally consider[s] high flow and low flow conditions". This suggests that these 3 KGE variants collectively emphasize high and low flows. . . is this at the expense of moderate flows?

P8 L20-25: This section reads more like methods than results. It would be helpful to begin the results section by presenting an overview of the most compelling findings. In Figure 6 caption, specify that Ehi is Kling-Gupta efficiency

P9 L6-7: "indicating that these combinations of SFCs are good candidates for general purpose hydrological studies..." True for these 33 watersheds, not necessarily in other locations. Specify that this conclusion is only for the study catchments. Also, while you're at it, you might point out that since all the calibration and evaluation scores were fairly high, this suggests that any of these objective functions (possibly except Elo) could do a reasonable job of predicting the overall hydrograph for these study catchments.

P9 L11-13: I don't see this in fig. 7. In (a) for Dfsh, it appears that Dinv produces the lowest Euclidean distance. In (b) for Dinv, it appears that Dfsh and Dall are nearly tied for the lowest Euclidean distance. In (c), yes it does appear that Dall has the lowest Euclidean distance for predicting for Dall.

P9 L16-17: Again, I don't see this in fig. 7. The two largest combinations of SFCs are Dfsh in (a) and Dall in (c). As stated, Ehi performs best for Dall in (c), but for Dfsh in (a), it appears that Eav performs better than Ehi. For the smallest combination of SFCs, Dinv, fig. 7 shows that Ehi performs best but the text states that Eav performs best. Please check that all panels of fig. 7 are labelled correctly and revise the text (or figure if needed) so that the figure and text are in agreement.

P9 L21 "measured by means of the standard deviation" would be better stated "measured by the standard deviation" so as not to confuse two different uses of the word "means".

P9 L22-23: "in addition to be predict well" Fix grammar.

P9 L32: There is no catchment 24003 presented in Fig 8a. Please correct.

P10 L13, definition of consistency: I found this definition a bit confusing and wonder if it

could be rephrased. Typically a ratio involves two variables to be compared, presented as "the ratio of X to Y". From the Fig. 9 caption it appears that values have been averaged across the 14 split-sample tests but it is unclear to me (and could be unclear to some readers) what the ratio represents. Ratio of "behavioral parameter sets that remain behavioural"... to what?

P10 L20-21: Text does not match fig. 9. Text states that consistency for Dfsh is 13%, whereas fig. 9 lists that value for Dinv and lists consistency for Dfsh as 0.309.

P10 L22: This does not seem "remarkable", and perhaps not even meaningful, given that there are only 3 data points (only 3 objective functions). With only 3 to compare, the association between greater consistency and greater numbers of SFCs could be attributable to chance. This also pertains to the conclusion you draw in discussion, P11 L9-10.

P10 L28: "model structure is not adequate for these catchments"... any ideas as to why? Do these catchments share any common attributes?

P11 L6: They're not really "three different sets of SFCs" because Dall contains the same SFCs as Dfsh and Dinv combined.

P11 L12: Arguably, SFCs can be hydrologically relevant (important) without necessarily being hydrologically representative or indicative "signatures" of the overall hydrograph.

P11 L19-20: "because they may not be key descriptors of the emergent hydrological processes at the catchment scale"... how does this square with results in Fig 6, in which you showed that performance of Dall and Dfsh was nearly equivalent to that of Ehi?

P12 L6-7: This doesn't fully fit with your findings, as you did not find that consistency was uniformly better for traditional objective functions than bespoke functions. Rather, Ehi had good consistency, whereas Elo had consistency poorer than two of the bespoke functions, and Eav had consistency that was roughly equivalent to two of the

bespoke functions. From your results (Fig 9) there is considerable variability in consistency values across traditional functions and also across bespoke functions. This also applies to your conclusions section P13 L19-20: "traditional objective functions select more consistently the same parameter sets as behavioural across the split-sample tests..." this is definitely true for Ehi (much greater consistency than all 3 bespoke functions), but not for Elo (lower consistency than 2/3 of bespoke functions), and is only marginally true for Eav (roughly equivalent to 2/3 bespoke functions, and more consistent than the third). My understanding of your results in Fig 9 are that they produce a somewhat mixed picture, with only (at best) weak support for your third hypothesis.

P12 L20-22: Can you clarify this section a bit? "may not be realistic for practical applications" [why not?] ... "might not be as critical for the stream ecology" [wouldn't that need to be determined empirically?] It's not clear to me what you're meaning to say here.

P12 L24-26: "Unless, ... both at once" Remove comma after "unless" and specify that "both" means both "ecologically relevant" and "hydrologically relevant" (although I dislike the latter term as explained above).

---

## Author Comment (AC1) · 29 Aug 2019

Dear Editor, Dear Reviewers,

We would like to thank you for your consideration of our manuscript and the careful review and feedback you provided us with. Please find below a detailed point-by-point answer to the reviewers' comments.

The main modifications in the revised manuscript include:
- Highlighting that the SFCs used in the tailored objective functions have not yet proven to be ecologically-relevant for the study catchments, and renaming the tailored objective functions to remove the misleading subscript including specific species;
- Replacing the objective function on average flows by the KGE on square-rooted flows;
- Providing an additional analysis on the performance of the six objective functions on the individual components of the original KGE criterion, as well as on the individual SFCs contained in the tailored objective functions;
- An extended discussion section to refocus on the main results of the study and also to include the limitations of this one.

A revised version of the manuscript can be found right after the point-by-point answers at the end of this document.

Thibault Hallouin, on behalf of the co-authors.

**Nomenclature:**
RXCY – Referee number X Comment number Y
AR – Authors' Reply

**Anonymous Referee #1**

**General comments**

**R1C1:** The differences in performance, stability, and consistency are evaluated in terms of KGEhi, KGEav, KGElo, Dinv, Dfsh, and Dall. These are all multi-objective criteria and their final value is an aggregated metric over multiple aspects of the hydrograph performance. Looking at the single components of KGE and also at single SFCs could potentially allow us to gain some insight into why differences in performance, stability, and consistency can be observed (e.g. is the performance of a particular bespoke objective function lower than the one of KGEhi because of a few SFCs that are very poorly predicted?). I think it is especially the "why" that helps us to learn more about the effect of objective functions on simulations and that ultimately helps us to improve predictions of various hydrograph aspects.

**AR:** We agree that a deeper analysis is useful to explore why the different objective functions may perform similarly or differently. We followed the reviewer's suggestions and analysed the performance on the different components of KGE (new section 4.6.1) and the single SFCs contained in the bespoke objective functions (new section 4.6.2). In addition, new section 4.6.3 considers the performance of the objective functions on 156 SFCs (including the 18 from new section 4.6.2) and 9 percentiles of the flow duration curve in order to look for trends in the performance of each objective function.

**R1C2:** Using the concept of consistency, this study looks at the ability of objective functions to consistently select the same set of behavioural parameters across multiple calibration time windows. Consistency is evaluated and compared across objective functions using a fraction-based score (i.e. fraction of parameter sets that is constantly behavioural). I think it could be interesting to also look at the parameter values themselves by e.g. plotting the values of the most sensitive parameters against model efficiency. Eventual patterns in parameter values might then be helpful to understand differences in model performance from different objective functions.

**AR:** This would represent an interesting additional analysis, however, this would require a whole manuscript by itself to satisfactorily cover the subject. Indeed, because of parameter interactions (10-parameter model used), 2D plots of parameter values against efficiency scores could lead to misinterpretation. Nonetheless, we provided the reader with some suggestion on how to extend/improve the consistency analysis in new section 5.3 on the limitations of the study (i.e. suggesting to use clustering analysis rather than dotty plots to further compare objective functions).

**R1C3:** Sometimes differences in model performance between objective functions are rather small (as mentioned by the authors in section 4.2). Is it possible that these differences only seem to be small because model performance cannot get too bad in the study catchments? It might be worth to compare results against a benchmark, such as a random selection of 1.

**AR:** Thank you for this interesting suggestion to use a benchmark. We opted for the use of a random selection of 1000 parameter sets, rather than one, in order to be consistent with the calibration strategy that identifies 1000 'behavioural' parameter sets. On all the relevant results plots (new Figures 5, 6, 8), the

results with the benchmark are added and referred under the fictional objective function 'R' (for random selection).

**R1C4:** The study is based on three hypotheses addressing model performance, stability, and consistency. The introduction provides the reader mostly the background and motivation for hypothesis H1 (model performance). However, there is not much information about the current knowledge and experience on the stability of model performance and the consistency in parameter selection. I recommend to extend the introduction so that it covers the whole range of questions addressed in the study.

**AR:** The literature on the comparison of objective functions for the prediction of SFCs with hydrological models has mostly been interested in the performance aspect, and rarely covered stability or consistency. However, by addressing **R1C6** and **R1C16** we reviewed new studies looking at model temporal robustness and consistency, which was added in the introduction: we covered the robustness (Garcia et al., 2017) and consistency (Visser-Quinn et al., 2019) aspects where relevant. We believe that this four-fold analysis represents a novel angle for the comparison of objective functions for the calibration of hydrological models for eco-hydrological applications.

**Specific comments**

**R1C5:** Abstract: More than half of the text in the abstract is introduction. I suggest to shorten that part and provide more information on the methods and results of the three hypotheses of the study (performance, stability, and consistency).

**AR:** We shortened the first half of the abstract that read more as introduction (as stated by the reviewer) and expanded on the second half with more details about the results and findings of the study.

**R1C6:** Introduction: The studies of Kiesel et al. (2017) and Pool et al. (2017) are often cited as examples for using streamflow characteristics in model calibration. However, there are many more studies using hydrological signatures for model calibration. Although these studies don't aim at simulating ecological flow indices it might be worth to cite some of them.

**AR:** Thank you for suggesting Hernandez-Suarez et al. (2018) and Zhang et al. (2016) in **R1C16**. We also added references to the studies by Mizukami et al. (2019), Garcia et al. (2017), and Visser-Quinn et al. (2019) in the introduction, and in the results sections whenever relevant to compare with our own findings.

**R1C7:** P4 L 4-14: It makes clearly sense to me that you select the same SFCs as have been used in previous studies. However, I am not sure about the use of the term "ecologically relevant" in e.g. your title. To my knowledge, the relevance of SFCs depends on the species of interest. I wonder if the same species are important (or exist) in Ireland as in Germany and the Southeastern US? If not, can the selected SFCs still be considered as ecologically relevant?

**AR:** Macro-invertebrates and fish communities (especially salmon and trout) are indeed very important for Irish rivers, and they support the delivery of key ecosystem services of Irish freshwater ecosystems (Feeley et al., 2017). However, we agree that the SFCs can only labelled as ecologically-relevant when empirical evidence can confirm that they are indeed the flow regime characteristics that Irish species are most sensitive to. Nonetheless, the challenges of predicting streamflow characteristics remain unchanged (ecologically-relevant or not), and hence we believe that the assumption does not impede on the

relevance of our case study. We emphasised this assumed "ecological relevance" in section 2.1: "*These three sets are assumed to be of some ecological relevance to the Irish study catchments for the purpose of comparing traditional and plausible tailored objective functions, however, empirical evidence is lacking to date to confirm whether they are the optimal indices for key invertebrate and fish species found in Irish rivers.*" (P4 L30-34). We argue that the design and findings of this study still address the challenges of making ecologically-relevant hydrological predictions, i.e. predicting specific streamflow characteristics, and as a result, we believe that the mention of ecological relevance in the title would allow the reader to clearly understand the scope of the research presented, and relate the work to complementary research already published in HESS (e.g. Pool et al. (2017), Visser-Quinn et al. (2019)).

**R1C8:** P4 L20: Seven years are selected for calibration. Are seven years enough to get a robust estimate of the SFCs used in this study? I think it would be worth to discuss that and think about the consequences for the results on the stability of performance.

**AR:** We agree with the reviewer that the length of the calibration period is important to consider and discuss. This is why we covered this in the new section 5.3 (P18 L22-31). In summary , we believe that this is sufficient for model calibration, because five years are enough to capture the hydrological variability (Merz et al., 2009), but because SFCs are used for calibration with the tailored objective functions, this is likely that the natural variability in SFCs for extreme flows is underestimated with a seven-year period.

**R1C9:** P4 L24-30: Can you say how many catchments are nested? How does that affect the generalization of the results? You mention that the catchments represent the diversity in soil types and geology - therefore, I would provide more explicit information about soil type and geology. Is there any snowfall in the study catchments?

**AR:** There are 15 distinct catchments, and as a consequence 18 nested catchments. This information is added in section 2.2: "*A total of 33 gauges (displayed on Figure 1b) featured sufficient data of good quality to be used as study catchments, amongst which 18 are nested within any of the 15 distinct catchments.*" (P5 L21-23) and complemented this by displaying both the 15 distinct catchments on new Figure 1a, and the 33 gauges on new Figure 1b.

**R1C10:** P5 L26: Model set-up: How was potential evapotranspiration calculated? How was the warming-up period selected in case of the moving split-sample tests?

**AR:** The potential evapotranspiration is provided by Met Éireann and calculated using the Penman-Monteith equation, this is now detailed in section 3.2 (P7 L7-9). The warm-up period of 5 years is applied prior the first full hydrological year displayed on new Figure A1. Details are added in section 3.2 (P7 L9-11). This is also added visually to Figure 3 on the rolling window split-sample method. For completeness, it is also added that the simulations are run for the whole period starting on the first day of the warm-up period until the last day of the 14$^{th}$ hydrological year (P7 L11-13), even if some years are not used. Then, the relevant hydrological years (that can be inferred by using information on Fig. A1 and Fig. 3 jointly), are the extracted from this series for each split-sample test.

**R1C11:** P6 L20: It is a common practice to not only transform flows to put more emphasize on low flows but also put more emphasize on mean flows (by e.g. calculating the sqrt of flows). What was the reason for calculating the mean of KGEhigh and KGElow instead of making a transformation?

**AR:** Yes, we agree with the reviewer that using the square-rooted flows is more satisfactory if one is interested in giving more importance on moderate flow conditions. Again, this was also suggested by the other reviewer. This is why we substituted the mean of KGE on high flows and KGE on low flows by the KGE of square-rooted flows for average flow conditions $E_{KG}{}^{Q\,0.5}$.

**R1C12:** P7 L12: Given that the focus of the paper is on the comparison of traditional and bespoke objective functions, I recommend to explicitly write down the equation for each of the three bespoke objective functions so that the reader doesn't have to check Table 2 to know which SFCs and how many are in each bespoke objective function.

**AR:** We understand the reviewer's point of view and the potential inconvenience for the reader, however, for reasons of clarity, we believe that writing down the Euclidean distance of a 13-SFC or a 18-SFC space does not produce a very readable equation, and we would like to keep this as is (i.e. a more generic Equation 4 with a reference to Table 1).

**R1C13:** P7 L12: The traditional objective functions (KGE variants) have their optimal value at 1 whereas the bespoke objective functions have an optimal value of 0. This can be very confusing when interpreting the results in Figs. 6, 7 and 8. I strongly recommend to define the bespoke objective functions in the same way as the KGE, i.e. 1 – Euclidean distance.

**AR:** Indeed, for consistency purposes, this is better to define the "tailored" (formerly "bespoke") objective functions as efficiency scores with an optimum at one (like KGE). This is now done, and Equation 4 was modified accordingly.

**R1C14:** P7 L16: Model evaluation: A description of the concepts of stability and consistency is missing in the section on model evaluation. On the other hand, many sections of the results (P8 L 20-25; P8 L27-P9 L2; P9 L 25-L31; P10 L9-15) start with an extended paragraph on methods. I would move these paragraphs to the section on model evaluation to make a clear separation between methods and results.

**AR:** There was indeed some overlap between the introduction of the results subsections and methods section. We have moved and extended these methodological aspects from the results subsections, to newly created subsections 3.4.1 to 3.4.5. The results subsections now only contain statements about the study results and comparison with similar studies when relevant.

**R1C15:** P8 L 10-14. How were the 14 combinations of the 7-year periods chosen?

**AR:** The 14 combinations were selected following the same approach as in de Lavenne et al. (2016). That is to say: from the 14 hydrological years at hand, to slide the window by one year from one sample-split to the next. This was preferred over more complicated boot-strapping strategies. This is clarified in in section 3.1: "*given the large dataset it would generate, it was decided to work only on 14 combinations by using the window of seven consecutive years, rather than more sophisticated boot-strapping strategies.*" (P7 L2-4).

**R1C16:** P11 L 4: There are more studies looking at the effect of objective functions on the prediction of streamflow characteristics than those of Vis et al. (2015), Kiesel et al. (2017), and Pool et al. (2017) (two examples are Hernandez-Suarez et al., 2018; Zhang et al., 2016)

**AR:** See **AR** to **R1C16** (addressed together).

**R1C17:** Discussion: About half of the discussion is about the implication of the results for climate change studies and studies on the prediction in ungauged basins. However, climate change and prediction in ungauged basins are not directly addressed in this study. I agree that the results are interesting and relevant for these two topics and it is important that you do discuss the implication of your results for climate change and prediction in ungauged basins. But generally, I recommend to rearrange the discussion to focus much more on the findings and limitations of this study: which hydrological processes are represented by different objective functions? What causes differences in performance and consistency? Why is there not much difference in the stability of model performance?

**AR:** This comment was found useful for both the results and discussion sections. Indeed, a more in-depth analysis of the various components of KGE and the "tailored" objective functions is carried out in the results section, which in turn helped to identify the strength of tailored and traditional objective functions (addressed in discussion sections 4.1 and 4.2). Moreover, new section 4.3 addressed the limitations of the study. However, it should be stressed here that the additional analyses did not fully uncovered the reasons for the difference in model consistency, beyond the assumption that this might be symptomatic of the problem of getting the right answer for the wrong reasons (Kirchner et al., 2006).

**Detailed comments**

**R1C18:** Title: Given that the results don't show very strong differences between performance and consistency between traditional and bespoke objective functions I would think about using a question mark at the end of the title.

**AR:** Agreed, a question mark was added in the title.

**R1C19:** P3 L29-32: H1: It is hypothesized that bespoke objective functions lead to a better model performance than traditional objective functions. This is a rather vague formulation and I would explicitly state that it is about a better model performance. H2: What is a "small" number of SFCs? Again, I would be more explicit in the formulation of the hypothesis.

**AR:** It appeared that the formulation of the hypotheses were rather confusing, therefore, we decided to reformulate these as the major objectives of the study instead, to help clarify the scope of the manuscript (P4 L14-23).

**R1C20:** Fig. 2: I agree that the information in Fig. 2 is interesting for someone working with the same data set. But its information is maybe not so relevant for the general reader. You could think about placing the figure in the supplemental material.

**AR:** We believe that the information contained in this figure is useful (especially in combination with new Figure 3 on the split-sample tests to be able to infer the exact years used in each catchment). However, in order to take into account the reviewer's suggestion, we have moved the figure to the appendix rather than the supplement (new Figure A1).

**R1C21:** Fig. 4: i) For me step f and g go hand in hand and I would merge them as you do with the calculation of the objective function in step c. Is there a reason why you use once the term "mean" and once the term "average"?

**AR:** The two steps have been merged in new Figure 4, following the reviewer's advice. The "mean" was used to refer to the summary across split-samples, while the "average" was used to refer to the summary across study catchments. However, during the revision of the manuscript, we noticed that the standard deviation is also used at the split-sample level in the definition of the performance stability. So new Figure 4 now refers to the more general term "statistics" in place of mean and average, and the new subsections 3.4.1 to 3.4.5 provide the details on what type of statistic is used in each case.

**R1C22:** P6 L30: I would suggest to shortly explain how exactly flows were normalized.

**AR:** Equations 5 and 6 were added in order to clarify this.

**R1C23:** Fig. 5: I am not sure if this figure is needed. The concept of the split sample test with a moving window is already well explained in the text.

**AR:** We believe it would be useful to keep it for clarity (see **R1C10**) and, given the additional information added to new Figure 3, it also gives the reader the opportunity to grasp the concept in a more visual manner, for those who better learn concepts this way.

**R1C24:** Fig. 7: Are the axis labels in 7a and 7b switched, i.e. should 7a be Dinv and 7b be Dfsh?

**AR:** Indeed, the labels were swapped by mistake. This is fixed in new Figure 6 (that replaces the old Figure 7).

**R1C25:** Fig. 7: You often use the term "Euclidean distance" when talking about the SFCs based objective functions. These objective functions are defined in section 3.2 and I don't think it is necessary to repeat that they are based on the Euclidean distance.

**AR:** Given the modifications in relation to **R1C13**, it becomes even irrelevant to refer to the "Euclidean distance". And mentions of the term have been removed in the revised manuscript.

**R1C26:** P9 L29: ". . . more holistic definition that traditional objective functions represent. . .". I think this is a delicate statement and needs some explanation, especially since KGE consists of three components and the bespoke objective functions consist of many more components.

**AR:** The redesigned analysis of the stability, following **R2C7**, moving from a visual to a more quantitative comparison (using the standard deviation across the split-sample tests) revealed that the marginal differences in stability are negligible when looking at averages across the study catchments, so that this piece of analysis becomes irrelevant. Nonetheless, we agree that the term "more holistic definition" would have required further explanation. For completeness: what was meant was that traditional objective functions consider every single day in the time series for model fitting in calibration, while some SFCs may only be looking at one day per month (e.g. ma26, the variability in March mean flow), i.e. less "holistic".

**R1C27:** P10 L27: I think it is rather the model performance that is consistent than the catchments themselves.

**AR:** For reasons of harmonisation between the analysis of the overall performance, the performance stability, the robustness, and the consistency, it was decided to not display the detailed consistency for each catchment individually. As a consequence, the paragraph the reviewer is referring to has been

removed. However, we agree that the phrasing was incorrect, and the consistency does refer to the model and not the catchment.

**Anonymous Referee #2**

**General comments**

**R2C1:** The second paragraph of the introduction is a bit generic in referring to "a range of streamflow characteristics". Some examples could help: low-flow versus high flow periods and their associated SFCs? Timing and duration statistics as opposed to magnitude statistics? It would help at this point in the introduction to give the reader a better sense of the types and categories of SFCs that are commonly used, and how and why different types of SFCs might be optimized using different functions. Different aspects of the flow regime are specifically named in methods (P4 L9-10), but would be better introduced and explained in the introduction.

**AR:** We agree with the review that mentioning the difference aspects of the flow regime that SFCs cover in required earlier in the text. We have added such description on P2 L5-L7: "*These SFCs describe specific aspects of the river flow regime that can be extracted from the streamflow hydrograph, and they can be categorised into magnitude, frequency, rate of change, timing, and duration of high, average, and low flow events (Poff et al., 1997)*".
Moreover, we have added further details for the generic part mentioned by the reviewer. However, due to some reorganisation in the introduction this can now be found in the 4th paragraph of the introduction (P2 L35-P3L4): "*Vis et al. (2015) found that certain combinations of objective functions fitted to flows, each focussing on different statistical aspects of the streamflow hydrograph (e.g. volume error, correlation) tend to be more suitable for the prediction of the magnitude of average flows, and the timing of moderate and low flows than using a single objective function fitted to flows, e.g. NSE alone.*".

**R2C2:** The adjective "bespoke" may be unfamiliar to some audiences, e.g. in the USA. Is there another term that could be more globally well understood? Perhaps "customized"?

**AR:** Thank you for raising this important issue to keep the manuscript understandable to everyone. We suggest replacing every mention of "bespoke" by "tailored" that we believe is more internationally understood.

**R2C3:** Also, can you provide a succinct definition of "behavioural" parameter and how parameters are identified as "behavioural"?

**AR:** Indeed, this was explained in old section 3.2 as "*the best 1% parameter sets are retained on the basis of their performance on the chosen objective function*" but there was no mention of the term behavioural. It is replaced in new section 3.3 by "*Eventually, in step (d), the best 1% parameter sets (i.e. those with the highest efficiency scores) are retained as "behavioural" on the basis of their performance on the chosen objective function, which yields a set of $10^3$ parameter sets.*" (see P8 L11-12 in the revised version).

**R2C4:** Fig. 2 caption indicates a substantial number of complete hydrological years were discarded from analysis (light blue) but this is not explained in methods. Why was this done? Was it so as to have an equal number of years of available data across catchments, and the minimum available was 14 years? Or

was this to facilitate the split-sample tests? Were there trade-offs in this decision, i.e. loss of precision due to using less calibration or validation data than was actually available?

**AR:** This is indeed to have an equal number of years across catchments. This is now mentioned: "*For catchments featuring more than 14 complete hydrological years, the additional available years were not used in order to avoid any positive or negative bias due to differences in data series length between these catchments and other catchments in the set.*" (P5 L16-18). And, there are indeed some trade-offs associated with this for the accuracy of the estimation of the observed SFCs, and this is discussed in the new section 5.3 on the study limitations (i.e. more uncertainty on the estimation of the observed SFCs).

**R2C5:** Throughout the text (e.g. P4 L8 and in Table 2), sets of SFCs are presented as belonging to "fish" or "invertebrates" when really they represent SFCs drawn from two sets of studies including Knight et al (2014) (fish) and Kakouei et al (2017) (invertebrates). That happened to use these taxonomic groups. As currently presented, this gives the impression that these sets of SFCs have empirically been demonstrated to be more important to fish than to inverts, or vice versa. Similarly, labeling the vectors and corresponding Euclidean distances as "Dfsh" and "Dinv" gives this impression. The cited studies from which these SFCs were derived were not focused on the relative importance of SFCs to one taxonomic group over another. To truly represent such broad taxonomic groups, studies across a wider set of geographic areas (not just the SE United States and Germany) and more diverse sets of species would be required. Regardless, this is not a goal of this paper. This could be presented differently, with the same SFCs and same analysis, in a way that would not give false impressions about overall taxonomic relevance for particular groups of SFCs.

**AR:** Thank you for raising this issue that could indeed mislead the reader. The assumption that these SFCs are ecologically-relevant is now clearly mentioned in section 2.1 P4 L30-34 (see also AR to **R1C7**): "*These three sets are assumed to be of ecological relevance to the Irish study catchments for the purpose of comparing traditional and plausible tailored objective functions, however, empirical evidence is lacking to date to confirm whether they are ecologically- relevant for key invertebrate and fish species found in Irish rivers.*". Moreover, the "tailored" (formerly "bespoke") objective functions were renamed in order to avoid confusion: $D_{inv}$ became $E_{SFC}^K$, Dfsh became $E_{SFC}^P$, and Dall became $E_{SFC}^{KP}$, with K standing for Kiesel et al. (2017) and P for Pool et al. (2017), to refer directly to the studies we relied on.

**R2C6:** There are some problems of disagreement between results text and results figures. For example, the results text in section 4.2 does not match the results shown in Fig. 7 (see specific comments below). Please ensure correct labeling of figure panels and revise text or figure as needed to ensure agreement. There is also disagreement between results text in section 4.4. and results shown in Fig. 9 (see specific comments below).

**AR:** There was a mistake in the labelling of Figure 7. This has been corrected in the new Figure 6 (replacing the old Figure 7). Apologies for the confusion.

**R2C7:** In the results as presented in section 4.3 and Fig. 8, stability of performance across the split-sample tests is apparently assessed only visually. Even for this study with a relatively small number of catchments (33) this becomes a bit unwieldy (and the reader is forced to search through 33 separate plots in each of 3 figure panels to find the examples mentioned in the text). If these methods were to be reproduced or adapted for another study area with a larger number of catchments (in the hundreds, as is common), then a figure such as Fig. 8 upon which the conclusions about stability rely would become untenable. This

paper would benefit from a more quantitative assessment of stability, presented perhaps as means and standard deviations across catchments instead of Fig. 8. Such a quantitative stability metric would (a) be much easier to understand and interpret, while presumably yielding the same results, namely that performance stability was not markedly different between traditional and bespoke objective functions, and (b) would be more repeatable and transferable to other studies in other regions, especially using large numbers of catchments.

**AR:** We replaced the qualitative (i.e. visual) assessment by quantitative assessment. To do so, as suggested by the reviewer, we used the standard deviation of the performance across the 14 split-sample tests. These new results are presented on Figure 6d-f that we believe are now easily reusable in studies featuring more catchments. The findings are the same, i.e. the stability is not significantly different with the six objective functions. However, due to the more holistic analysis, the detail catchment per catchment is lost, and the counter-examples previously mentioned are now absent.

**R2C8:** In conclusions, I'm not sure that hypothesis 3 is strongly supported by results in Fig 9 (see specific comments below).

**AR:** Addressed in **AR**'s to **R2C43.**

**R2C9:** Table 2 is cited before Table 1 (section 2.1 and 2.3, respectively). Likewise, Fig. 1 is cited after Figs 2 and 3. Please ensure all tables and figures are cited in proper order.

**AR:** Reordering done for all tables and figures.

**Specific comments**

**R2C10:** P2 L5: "The prediction of streamflow conditions" [need to add or specify "at ungauged locations"]

**AR:** Done (P2 L7).

**R2C11:** P2 L7 "hydrological models that produce streamflow hydrographs" [may want to specify these are simulated hydrographs as opposed to observed hydrographs]

**AR:** Thank you, done (P2 L11).

**R2C12:** P2 L15 "across a range of streamflow characteristics" … does this particularly involve high-flow vs low-flow periods?

**AR:** Already addressed in **AR**'s to **R2C1**.

**R2C13:** P2 L16-17: sentence fragment. Delete "although"?

**AR:** Deleted (P3 L4).

**R2C14:** P3 L9: is there a missing comma? "captured in the effective parameter values of the model, could be compromised…"

**AR:** There is indeed. Added (P3 L35).

**R2C15:** P3 L12-12: "... relating to its own preferences for living conditions" this was also demonstrated by Knight et al (2014).

**AR:** Thank you. Added (P4 L4).

**R2C16:** P3 L13-14: "when several species are considered simultaneously, the number of SFCs to predict will increase accordingly" ... yes, in general, but that depends on the habitat requirements of the species in question. Species with similar behavior (eg foraging strategies), reproductive timing, and physical (eg thermal) niches may actually share a common set of SFCs as being most relevant to them.

**AR:** Thank you for this contribution, it now reads as: "*When several species are considered simultaneously, the number of SFCs to predict is likely to increase, even though some stream species may respond to broadly similar streamflow characteristics.*". (P4 L8-9)

**R2C17:** P4 L3-7: Any evidence that these SFCs are ecologically relevant in the Irish catchments?

**AR:** There is evidence that fish communities (salmon and trout in particular) and macro-invertebrates (e.g. Mayfly) are essential for the delivery of key ecosystem services in Irish freshwater ecosystems (Feeley et al., 2017). However, there is no empirical evidence to back up that the SFCs used by Pool et al. (2017) and Kiesel et al. (2017) are the ones these key species are sensitive to in Irish rivers (see **R1C7** for how this was addressed in the manuscript).

**R2C18:** P4 L8: "Only two hydrological indices are common between the two communities' respective streamflow preferences". I don't think based solely on the cited studies in the Southeastern US and Germany that such sweeping statements can be made about streamflow preferences across such broad taxonomic groups (fish and invertebrates). More accurately, this is a comparison between two sets of studies on two different continents that happened to use different taxonomic groups in their analysis. It does not support conclusions about which SFCs matter most to which taxonomic groups outside the respective study areas of the cited studies. I am familiar with the research in the Southeastern US (Knight et al 2014) which did not use any invertebrate data at all, and so should not be used to suggest that certain SFCs are more (or less) important to fish relative to inverts in that study region.

**AR:** We believe that the AR's to **R2C5** (i.e. renaming of the tailored objective functions) and to **R1C7** (i.e. explicitly mentioning the ecological relevance is an assumption for our study region, due to the lack of evidence as of now) address this.

**R2C19:** P4 L13: "calculations were vectorised" ... can you explain a bit more what this means? I'm familiar with EflowStats but not Python. Perhaps Python users know exactly what you mean, but a brief explanation could help readers with less familiarity.

**AR:** We expanded on this: "*However, all computations for this study were carried out in Python where the NumPy package was used to vectorise the calculations of the SFCs (i.e. to formulate the calculations as arithmetic operations between vectors and matrices). This makes use of algorithms directly coded in C, avoiding the redundant interpretation of Python statements in loops which is significant for iterations over a very large number of streamflow time series (i.e. $8.316 \times 10^6$) (Hallouin, 2019a).*" (P5 L3-7).

**R2C20:** In Table 2, the column header "target species" (fish or invertebrates) might give a casual reader the impression that these SFCs were empirically deemed important in the study-area catchments. Rather, they were gleaned from Knight et al (2014) (fish) or from Kakouei et al (2017) (invertebrates). More importantly, it could give a false impression that these SFCs are generally more important to the taxonomic group listed which is not necessarily the case based on the previous studies cited. This could be clarified in the table caption or the accompanying text, or by changing "target species" to "citation" and listing the corresponding paper from which the SFC was obtained.

**AR:** We removed the column mentioned by the reviewer in Table 1 (formerly Table 2) in order to avoid implying any proven relation between taxonomic groups and the SFCs in our study catchments.

**R2C21:** Also in Table 2, table caption says "SFC" but column header says "indicator". Would be better to use one term consistently for clarity. This goes for the main text as well, where "SFCs" and "hydrologic indices" are both used.

**AR:** We replaced "indicator" by "SFC" in the table. Moreover, we replaced "indicator" / "indices" in text by "SFC" or simply "characteristic". Exceptions: in the introduction before SFC was defined, and in the discussion section 5.1 where ecologically-relevant SFCs are compared with hydrological signatures (because the actual aspect covered by an SFC/signature can actually be the same).

**R2C22:** P4 L20: "in order to have at least seven years for calibration and seven years for evaluation. . ." Can you justify (perhaps with citation) why seven years is an acceptable POR for calibration/validation?

**AR:** We referred to the study by Merz et al. (2009) who found that five years were enough to capture the hydrological variability in catchments (P5 L13-14). However, we also mentioned the potential impacts of using a relatively short period in the discussion section 5.3 on the limitations of the study (P18 L22-31).

**R2C23:** P4 L27-28: "hence representing a good sample of the diversity of Irish soils and geology" Except that from Fig 3 it appears that higher-elevation or mountainous catchments were not well represented...?

**AR:** Indeed, there is a lack of catchments in the Wicklow mountains (East coast) and the mountainous edge along the Atlantic coast. The statement was softened: "*They are located throughout the country, and they represent a diversity of Irish soils and geology (Figure 1c,d), despite lacking some of the most elevated catchments in the Wicklow mountains (relief on the East coast), and the mountainous edge on the Atlantic coast (Figure 1b).*" (P5 L24-26), while additional maps on Figure 1 (formerly Figure 3) are added to display the main geological regions and soil groups, we also provide more information on the catchments in new Table A1.

**R2C24:** Fig 1 and Fig S1 are essentially redundant to each other, and Table 1 and Table S3 are redundant, except for slight differences in parameter presentation. Parameter representations in Table S3 appear to match Fig S1 but not Fig 1. Suggest retaining only one version of this figure/table combination, or if repeating in the supplement then ensure consistent parameter representations throughout.

**AR:** We harmonised this so that parameters match (kept the figure originally present in the supplement for both). We would like to keep the figure duplicated in the supplement material for convenience for the reader going through the detailed description of the model equations provided there.

**R2C25:** P5 L14 and L19: How are energy-limited and water-limited periods defined?

**AR:** We have added this detail: "*In water-limited periods (i.e. when potential evapotranspiration exceeds incident rainfall)*" (P6 L15-16). Energy-limited refers to the energy available for evapotranspiration and applies "*when potential evapotranspiration is less than incident rainfall*" (P6 L10-11).

**R2C26:** P5 L27: What formulation of ET is this? E.g. Penman-Monteith?

**AR:** Yes, the data provided by Met Éireann is using the FAO Penman-Monteith formula. See P7 L7-9: "*The potential evapotranspiration is calculated by Met Éireann using the FAO Penman-Monteith formula (Allen et al., 1998) with coefficients adjusted for Irish conditions and meteorological data at their synoptic weather stations.*".

**R2C27:** P6 L23: do these KGE variants really encompass "the entirety of the observed and simulated hydrographs"? It seems the first KGE emphasizes high flows, the second (inverted) emphasizes low flows, and the third "equally consider[s] high flow and low flow conditions". This suggests that these 3 KGE variants collectively emphasize high and low flows. . . is this at the expense of moderate flows?

**AR:** Yes, this was not satisfactory to use the mean of $KGE_{hi}$ and $KGE_{lo}$ as an objective function for average flow conditions. In the revised version, $KGE_{av}$ is replaced by KGE on square-rooted flows to put more emphasis on moderate flow conditions.

**R2C28:** P8 L20-25: This section reads more like methods than results. It would be helpful to begin the results section by presenting an overview of the most compelling findings.

**AR:** We have made sure that the results section only contains results now, and the methodological aspects, including the one provided here are moved to the methods sections 3.4.1 to 3.4.5.

**R2C29:** In Figure 6 caption, specify that Ehi is Kling-Gupta efficiency

**AR:** We agree that Ehi was not very explicit for a reader glancing at the Figure. Because, we decided to rename all objective functions to $E_{KG}$, including the traditional objective functions so that their name is more explicit: $E_{hi}$ becomes $E_{KG}^{Q}$, $E_{av}$ becomes $E_{KG}^{Q\,0.5}$, and $E_{lo}$ becomes $E_{KG}^{Q\,-1}$. And we have added the explanation for $E_{KG}^{Q}$ in the caption of new Figure 5 (replacing old Figure 6)

**R2C30:** P9 L6-7: "indicating that these combinations of SFCs are good candidates for general purpose hydrological studies. . ." True for these 33 watersheds, not necessarily in other locations. Specify that this conclusion is only for the study catchments. Also, while you're at it, you might point out that since all the calibration and evaluation scores were fairly high, this suggests that any of these objective functions (possibly except Elo) could do a reasonable job of predicting the overall hydrograph for these study catchments.

**AR:** We have reformulated and added this precision: "*indicating that all six objective functions are useful to find parameter sets representative of the hydrological behaviour of our catchments.*" (P12 L8-9). The fact that all these objective functions perform well is mentioned prior this, with the help of the newly introduced benchmark (following suggestion in **R1C3**): "*On average, all six objective functions perform well in reproducing the observed hydrograph when more weight is given to predicting high flows well, with $E_{KG}^{Q}$ scores in evaluation between 0.69 and 0.82 in evaluation (Figure 5a). They largely outperform the average benchmark score of 0.40*" (P12 L6-8).

**R2C31:** P9 L11-13: I don't see this in fig. 7. In (a) for Dfsh, it appears that Dinv produces the lowest Euclidean distance. In (b) for Dinv, it appears that Dfsh and Dall are nearly tied for the lowest Euclidean distance. In (c), yes it does appear that Dall has the lowest Euclidean distance for predicting for Dall.

**AR:** As per comment **R2C6** and **R1C24**, there was a mistake in the labelling due to a late reordering of the objective functions from the one with the smallest number of SFCs, to the one with the largest number, that was not replicated in the label ordering. This is now fixed in the new Figure 6 (replacing old Figure 7).

**R2C32:** P9 L16-17: Again, I don't see this in fig. 7. The two largest combinations of SFCs are Dfsh in (a) and Dall in (c). As stated, Ehi performs best for Dall in (c), but for Dfsh in (a), it appears that Eav performs better than Ehi. For the smallest combination of SFCs, Dinv, fig. 7 shows that Ehi performs best but the text states that Eav performs best. Please check that all panels of fig. 7 are labelled correctly and revise the text (or figure if needed) so that the figure and text are in agreement.

**AR:** Please see AR's to **R2C31**.

**R2C33:** P9 L21 "measured by means of the standard deviation" would be better stated "measured by the standard deviation" so as not to confuse two different uses of the word "means".

**AR:** Thank you, this was replaced (P13 L10).

**R2C34:** P9 L22-23: "in addition to be predict well" Fix grammar.

**AR:** Thank you, this was corrected: "*in addition to predict well on average*" (P13 L11-12).

**R2C35:** P9 L32: There is no catchment 24003 presented in Fig 8a. Please correct.

**AR:** Thank you for spotting the typo, it should have read as 34003. The detailed analysis per catchment was, however, replaced by a more quantitative analysis using the standard deviation (as recommended in **R2C7**) so that nothing needed to be fixed in the revised text.

**R2C36:** P10 L13, definition of consistency: I found this definition a bit confusing and wonder if it could be rephrased. Typically a ratio involves two variables to be compared, presented as "the ratio of X to Y". From the Fig. 9 caption it appears that values have been averaged across the 14 split-sample tests but it is unclear to me (and could be unclear to some readers) what the ratio represents. Ratio of "behavioral parameter sets that remain behavioural" ... to what?

**AR:** Refinement on the explanation for the concept of consistency is provided in the new section 3.4.4 : "*The consistency is calculated from the ratio of the number of model parameter sets identified as behavioural that are common to all 14 split-sample tests divided by the total number of behavioural parameter sets (i.e. $10^3$), and then the average ratio across the 33 study catchments is calculated in subsection 4.5 to obtain the model consistency with a given objective function used to identify the behavioural parameter sets. The consistency ranges from zero to one, with an optimal value at 1.*" (P11 L8-11).

**R2C37:** P10 L20-21: Text does not match fig. 9. Text states that consistency for Dfsh is 13%, whereas fig. 9 lists that value for Dinv and lists consistency for Dfsh as 0.309.

**AR:** Again, there was an error in the labelling (see **AR**'s to **R2C31**). Apologies renewed.

**R2C38:** P10 L22: This does not seem "remarkable", and perhaps not even meaningful, given that there are only 3 data points (only 3 objective functions). With only 3 to compare, the association between greater consistency and greater numbers of SFCs could be attributable to chance. This also pertains to the conclusion you draw in discussion, P11 L9-10.

**AR:** We agree that the term "remarkable" is inappropriate and this aspect is now only mentioned as a remark, and a suggestion for further research on this aspect is provided: "*Moreover, the number of SFCs contained in the tailored objective function is another aspect that may need to be considered, given that the consistency seems to improve with the number of SFCs the objective function contains. However, only three set of SFCs were tested in this study, and more research would be required to confirm this hypothesis.*" (P17 L32-35).

**R2C39:** P10 L28: "model structure is not adequate for these catchments" ... any ideas as to why? Do these catchments share any common attributes?

**AR:** The catchments in question (15005 and 34024) do not appear to share similar geologies or soil types (see new Figure 1). Their baseflow indices, i.e. 0.71 and 0.52, respectively, are not peculiar in the set of 33 catchments (new Table A1). They are relatively small (380 $km^2$ and 128 $km^2$, respectively), but other smaller catchments perform better on consistency (new Table A1). So, to our knowledge, there is no obvious common feature that would be peculiar compared with the other catchments to explain this pattern. However, given that we decided to harmonise the level of analysis to average across the study catchments in order to be reproducible in future studies working with a large number of catchments, this anomaly in the consistency for these two catchments is not available in the new version of the manuscript.

**R2C40:** P11 L6: They're not really "three different sets of SFCs" because Dall contains the same SFCs as Dfsh and Dinv combined.

**AR:** This is true. The term "different" was removed in the revised version. "*This study confirmed these separate findings using the same set of SFCs in Irish study catchments.*" (P17 L6).

**R2C41:** P11 L12: Arguably, SFCs can be hydrologically relevant (important) without necessarily being hydrologically representative or indicative "signatures" of the overall hydrograph.

**AR:** We originally used "hydrologically-relevant" because signatures and SFCs could be considered as the same thing (i.e. an indicator extracting some information from a hydrograph), but we understand the reviewer's point of view, and this adjective is probably misleading. We reformulated the section paragraph of section 5.1 (P17 L10-28) to remove any mention of "hydrologically-relevant" and used more explicit descriptions in place. Same applies to the conclusion (see **R2C45**).

**R2C42:** P11 L19-20: "because they may not be key descriptors of the emergent hydrological processes at the catchment scale" ... how does this square with results in Fig 6, in which you showed that performance of Dall and Dfsh was nearly equivalent to that of Ehi?

**AR:** The difference in ranking across the six objective functions whether they are compared on performance or on consistency is in fact at the origin of this is an assumption. Indeed, $E_{SFC}^{KP}$ (formerly $D_{all}$)

and $E_{SFC}^P$ (formerly $D_{fsh}$) show good performances on KGE. But given that Euser et al. (2013) found that their set of hydrological signatures yielded improvements on model consistency, it is hypothesised here (conditional tense) that this might be the reason why they show lower model consistency. As stated afterwards: *"this may be symptomatic of the problem of getting the right answer with a model for the wrong reasons (Kirchner, 2006)"* (P17 L19-20).

**R2C43:** P12 L6-7: This doesn't fully fit with your findings, as you did not find that consistency was uniformly better for traditional objective functions than bespoke functions. Rather, Ehi had good consistency, whereas Elo had consistency poorer than two of the bespoke functions, and Eav had consistency that was roughly equivalent to two of the bespoke functions. From your results (Fig 9) there is considerable variability in consistency values across traditional functions and also across bespoke functions. This also applies to your conclusions section P13 L19-20: "traditional objective functions select more consistently the same parameter sets as behavioural across the split-sample tests. . ." this is definitely true for Ehi (much greater consistency than all 3 bespoke functions), but not for Elo (lower consistency than 2/3 of bespoke functions), and is only marginally true for Eav (roughly equivalent to 2/3 bespoke functions, and more consistent than the third). My understanding of your results in Fig 9 are that they produce a somewhat mixed picture, with only (at best) weak support for your third hypothesis.

**AR:** Indeed, the consistency of $E_{av}$ and $E_{lo}$ (now $E_{KG}^{Q-1}$) was not very good. By replacing $E_{av}$ by KGE on square-rooted flows ($E_{KG}^{Q\,0.5}$), its consistency was largely improved (see new Figure 7). The objective functions featuring more emphasis on low flow conditions are the least consistent ones. The results section on consistency was rephrased, in particular: *"The two objective functions yielding the lowest consistencies are $E_{KG}^{Q-1}$ and $E_{SFC}^K$, with 0.19 and 0.13, respectively. The main similarity between these two objective functions is that they focus more on low flow conditions than average or high flow conditions. This may be the reason for their lack of consistency."* (P14 L6-8). Moreover, the binary comparison originally made between traditional objective functions and bespoke objective functions is relaxed: *"However, the consistency analysis revealed that the sample of parameter sets found suitable in calibration are less consistent across different split-sample tests with this type of objective function than with two of the traditional objective functions (i.e. $E_{KG}^Q$ and $E_{KG}^{Q\,0.5}$)."* (P17 L6-9).

**R2C44:** P12 L20-22: Can you clarify this section a bit? "may not be realistic for practical applications" [why not?] . . . "might not be as critical for the stream ecology" [wouldn't that need to be determined empirically?] It's not clear to me what you're meaning to say here.

**AR:** In the original manuscript, what was meant was that the sign of the deviation from the observed SFC is not taken into account in an objective function relying on the absolute error, and it was assumed that if one has to decide between parameter sets underestimating or overestimating low flows, one may want to favour parameter sets underpredicting in low flows as a conservative measure. But it is agreed that some of this would need to be determined empirically. The second aspect had to do with the normalisation of the deviation from the observed SFC that is used in the objective function, and one may want to include a weight reflecting the relative predictive power of a given SFC on the presence/absence, or the number of the target species (recently used by Visser-Quinn et al., 2019). The second point is retained in the revised version because empirical evidence (i.e. see Visser-Quinn et al., 2019) exists, while the first point is removed from the discussion.

**R2C45:** P12 L24-26: "Unless, . . . both at once" Remove comma after "unless" and specify that "both" means both "ecologically relevant" and "hydrologically relevant" (although I dislike the latter term as explained above).

**AR:** The comma was removed, and the mention of "hydrologically-relevant" is not used, as recommended in **R2C41**: "*This study shows that a gain in fitting performance for the SFCs may hide a loss in consistency in the behavioural parameter sets across the split-sample tests. This highlights that fitting ecologically-relevant SFCs well is not necessarily a guarantee of representing all the key hydrological processes (i.e. informative signature) defining the catchment response. Unless streamflow characteristics are proven to be both ecologically-relevant and an informative signature at once, carefully selected traditional objective functions fitted to flows are likely to remain preferable to predict ecologically-relevant streamflow predictions to avoid consistency issues.*" (P21 L7-12).

[revised manuscript text omitted]

---

## Referee Report (RR1)

**Review of the manuscript „Calibration of hydrological models for ecologically-relevant streamflow predictions: a trade-off between performance and consistency?" by Hallouin et al.**

Dear editor and authors,

I thank the authors for their well-structured answers and for rigorously answering and addressing many of the reviewers comments. I especially appreciate that some major concerns have been addressed, such as a more detailed analysis of the three KGE components and single SFCs, the introduction of a lower benchmark, restructuring the methods chapter, and the increase of study-specific information in the introduction. I very much enjoyed reading this version of the manuscript and I personally think this is a study of good quality with a sound and novel modelling approach and nice figures supporting the conclusion. I only have a few minor comments left that could be addressed in the review.

**Comments**

P1 L2: I am not sure if "In catchments with little human-induces hydro-morphological changes" is needed, because there are hydrological model that use e.g. land use as input and should be able to deal with changes.

P4 L15-23: To me this paragraph is like a short summary of the methods. I preferred the previous version with the hypotheses. Having hypotheses or questions helps to guide the readers focus on specific aspects in the results.

P5 L20: Could you add a sentence explaining the principle behind the quality check?

P5 L 30 (Rainfall runoff model): Is the model used in a lumped or semi-distributed way? I was wondering if you have any snowfall in the 33 catchments and if the model has the option to simulate snow accumulation and snowmelt?

P6 L5: I don't think the statement "…needed for an EPA funded project Pathways…" is relevant for the reader.

P6 L 24-27: I don't fully understand what the difference between the studies of de Lavenne et al. (2016) and Coron et al. (2012) is. Could you reformate the sentences that to make it more clear?

P11 L1: I would suggest to extend to section title to "Consistency in the selection of model parameter values" to make clear that consistency is about the parameters whereas stability and robustness are about performance.

P13 L29-30: The statement that tailored objective functions suffer more from overfitting than traditional objective function is based on Figure 6. However, to make a fair comparison of the robustness, the robustness of the traditional objective functions should be calculated by calculating the difference between a traditional objective function (and not the ESFCs) in calibration and validation when calibrated on the traditional objective function.

P15 L27: I couldn't find the evaluation of model simulations on 156 SFCs in the methods section. I recommend to add it somewhere to have a complete description of the evaluation in the methods section.

---

## Author Response (AR2)

Dear Editor, Dear Reviewers,

Thank you for your valuable comments on our manuscript. We believe that the comments of the reviewers helped us in further improving the quality of the manuscript. Please find below a detailed point-by-point answer to the reviewers' comment.

Thibault Hallouin, on behalf of the co-authors.

**Nomenclature**
RXCY – Referee number X comment number Y
AR – Authors' Reply

**Anonymous Referee #1**

**R1C1:** P1 L2: I am not sure if "In catchments with little human-induces hydro-morphological changes" is needed, because there are hydrological model that use e.g. land use as input and should be able to deal with changes.

**AR:** We agree that some hydrological models where land cover is used to infer some parameter values can be used in those instances (Note: this is not the case with the model used in this study). This was rephrased as "*In catchments with little human-induced alterations of the flow regime (e.g. abstractions, regulations)*" [P1 L2].

**R1C2:** P4 L15-23: To me this paragraph is like a short summary of the methods. I preferred the previous version with the hypotheses. Having hypotheses or questions helps to guide the readers focus on specific aspects in the results.

**AR:** This section was restructured using research questions as suggested by the reviewer [P4 L8-22]. The conclusions were restructured as well in order to provide answers to those questions [P20 L13-20].

**R1C3:** P5 L20: Could you add a sentence explaining the principle behind the quality check?

**AR:** The quality of data from each hydrometric station was assessed by considering the type of station and the quality of the rating relationship established for it using spot gauging measurements. The number of such measurements, the goodness of fit of the rating equation to these measurements and their coverage of low and high flow extremes were considered in the evaluation. The sentence reads now as: "*Catchment selection was also influenced by the quality of the discharge data, including the goodness of fit of the rating equation at the gauge, the number of measurements, and their coverage of low flow and high flow extremes*" [P5 L21-22].

**R1C4:** P5 L 30 (Rainfall runoff model): Is the model used in a lumped or semi-distributed way? I was wondering if you have any snowfall in the 33 catchments and if the model has the option to simulate snow accumulation and snowmelt?

**AR:** The model is used in a lumped manner and does not contain a snow module, as snowfall is negligible under Irish conditions. This information was added in section 2.3: "*The model does not contain any snow component as it is infrequent in Ireland*" [P6 L18-19], and in section 3.2: "*The SMART model is used in a lumped manner to predict streamflow at the catchment outlet*" [P7 L6] .

**R1C5:** P6 L5: I don't think the statement "...needed for an EPA funded project Pathways..." is relevant for the reader.

**AR:** Removed [P6 L4].

**R1C6:** P6 L 24-27: I don't fully understand what the difference between the studies of de Lavenne et al. (2016) and Coron et al. (2012) is. Could you reformate the sentences that to make it more clear?

**AR:** This section was expanded: "*Split-sample tests are commonly used to analyse the performance of hydrological models, where the study period is broken down into two periods, one for calibration and one for evaluation (Klemes, 1986). Coron et al. (2012) proposed a generalised split-sample test using a sliding window of a given duration across the study period: calibration is carried out on the given window, and the model performance is evaluated for all other independent windows in the study, thus evaluating on more than one period. De Lavenne et al. (2016) simplified this approach to evaluate on all data not included in the window (i.e. the years before and/or after the sliding window), thus evaluating on one period only*" [P6 L23-29].

**R1C7:** P11 L1: I would suggest to extend to section title to "Consistency in the selection of model parameter values" to make clear that consistency is about the parameters whereas stability and robustness are about performance.

**AR:** Section was renamed "*Consistency in the selection of the model parameter values*" [P10 L15].

**R1C8:** P13 L29-30: The statement that tailored objective functions suffer more from overfitting than traditional objective function is based on Figure 6. However, to make a fair comparison of the robustness, the robustness of the traditional objective functions should be calculated by calculating the difference between a traditional objective function (and not the ESFCs) in calibration and validation when calibrated on the traditional objective function.

**AR:** We agree that Figure 6 only partially supports the statement. We rephrased this part to suggest overfitting as a possible explanation of this behaviour: "*This difference in robustness may be caused by tailored objective functions suffering more from overfitting than the traditional objective functions*" [P13 L12-13].

**R1C9:** P15 L27: I couldn't find the evaluation of model simulations on 156 SFCs in the methods section. I recommend to add it somewhere to have a complete description of the evaluation in the methods section.

**AR:** Thank you for highlighting this gap in the methodology section, the sub-sub-section 3.4.6 "*Analysis on the performance on a large set of SFCs*" was added [P11 L12-17].

**Anonymous Referee #1**

**R2C1:** My major concern is that the text (in all sections, from abstract to conclusions) needs to be re-shaped/shortened in a way that it is more clear.
In its present form the paper is (unnecessarily) verbose and convoluted.
I am sure that if the text is reviewed, the paper will be largely improved.

**AR:** The revised version was revised in view to remove unnecessary sentences in order to provide a clearer and more concise text.

**R2C2:** I also think the title needs some thought.
The way in which "performance" and "consistency" is in the title might lead to confusion for a reader.

**AR:** We believe that the confusion mentioned here is also related to the issue raised in **R1C7**. We propose the amended title: "*Calibration of hydrological models for ecologically-relevant streamflow predictions: a trade-off between fitting well to data and estimating consistent parameter sets?*"

**R2C3:** Also, I miss some updated references in the section 5.

**AR:** We have checked all references in section 5, and we could not find any missing reference.

[revised manuscript text omitted]

---

## Author Response (AR3)

Dear Prof. Jan Seibert (Editor), Dear Reviewer,

We would like to thank you for your careful review of our manuscript, and we are grateful for your final acceptance for publication in HESS.

On behalf of all authors,

Kind regards,
Thibault Hallouin